# Beyond CVaR: Leveraging Static Spectral Risk Measures for Enhanced Decision-Making in Distributional Reinforcement Learning

**Mehrdad Moghimi** [1]   **Hyejin Ku** [1]

## Abstract

In domains such as finance, healthcare, and robotics, managing worst-case scenarios is critical, as failure to do so can lead to catastrophic outcomes. Distributional Reinforcement Learning (DRL) provides a natural framework to incorporate risk sensitivity into decision-making processes. However, existing approaches face two key limitations: (1) the use of fixed risk measures at each decision step often results in overly conservative policies, and (2) the interpretation and theoretical properties of the learned policies remain unclear. While optimizing a static risk measure addresses these issues, its use in the DRL framework has been limited to the simple static CVaR risk measure. In this paper, we present a novel DRL algorithm with convergence guarantees that optimizes for a broader class of static Spectral Risk Measures (SRM). Additionally, we provide a clear interpretation of the learned policy by leveraging the distribution of returns in DRL and the decomposition of static coherent risk measures. Extensive experiments demonstrate that our model learns policies aligned with the SRM objective, and outperforms existing risk-neutral and risk-sensitive DRL models in various settings.

## 1. Introduction

In traditional Reinforcement Learning (RL), the goal is to find a policy that maximizes the expected return (Sutton & Barto, 2018). However, considering the variations in rewards and addressing the worst-case scenarios are critical in some fields such as healthcare or finance. A risk-averse policy can help address the reward uncertainty arising from the stochasticity of the environment. This risk aversion can stem from changing the objective from expectation to other risk measures such as the Conditional Value-at-Risk (Bäuerle & Ott, 2011), coherent risk measures (Tamar et al., 2017), convex risk measures (Coache & Jaimungal, 2023), and Entropic-VaR (Ni & Lai, 2022). Another approach is limiting the worst-case scenarios by using constraints such as variance (Tamar et al., 2012) or dynamic risk measures (Chow & Pavone, 2013) in the optimization problem.

Another area of research that has gained attention for risk-sensitive RL (RSRL) is Distributional RL (DRL) (Morimura et al., 2010; Bellemare et al., 2017). This paradigm diverges from the traditional RL by estimating the return distribution instead of its expected value. DRL algorithms not only demonstrate notable improvements compared to conventional RL methods but also enable a variety of new approaches to risk mitigation. In this context, a few risk measures such as CVaR (Stanko & Macek, 2019; Keramati et al., 2020), distortion risk measure (Dabney et al., 2018a), Entropic risk measure (Liang & Luo, 2024), or static Lipschitz risk measure (Chen et al., 2024) have been explored.

In the DRL framework, applying a fixed risk measure at each step leads to policies that are optimized for neither static nor dynamic risk measures (Lim & Malik, 2022). In this case, action selection at different states are not necessarily aligned with each other, which can lead to policies that are sub-optimal with respect to the agent's risk preference. This issue, known as time inconsistency, is a common challenge in risk-sensitive decision-making (Shapiro et al., 2014). Intuitively, this misalignment can be understood as the fact that finding optimal policies starting from different states can yield different and inconsistent policies. To mitigate this, dynamic risk measures were introduced (Ruszczyński, 2010), which evaluate risk at each time step, unlike static risk measures that assess risk over entire episodes. However, dynamic risk measures are difficult to interpret, limiting their practical applicability (Majumdar & Pavone, 2020; Gagne & Dayan, 2022).

Optimizing static risk measures is more interpretable since it can be described as finding the policy that gives the best possible outcome in the worst-case scenario. However, unlike dynamic risk measures, the risk preference that the policy optimizes for at later stages is unclear. Traditionally, calcu-

---

[1]Department of Mathematics and Statistics, York University, Toronto, Canada. Correspondence to: Hyejin Ku <hku@yorku.ca>.

*Proceedings of the 42nd International Conference on Machine Learning*, Vancouver, Canada. PMLR 267, 2025. Copyright 2025 by the author(s).

lating these risk preferences has been limited to CVaR due to computational complexity (Bäuerle & Ott, 2011; Bellemare et al., 2023). However, we demonstrate that by leveraging the decomposition of coherent risk measures (Pflug & Pichler, 2016) and the return distribution within the DRL framework, these evolving risk preferences can also be computed for more general spectral risk measures. The intuition behind this approach is that a decision maker selects an initial risk preference, but it may change as new information becomes available over time. It is important to emphasize that, unlike previous works (Chow et al., 2015; Stanko & Macek, 2019) that use the decomposition of CVaR to derive optimal policies, we utilize this decomposition only to explain the behavior of the optimal policy, not for policy optimization. In fact, Hau et al. (2023) have demonstrated that the decomposition of coherent risk measures cannot be reliably applied for policy optimization, and the optimality claims in those works are inaccurate.

The contributions of our work are as follows:

- We propose a novel DRL algorithm with convergence guarantees that optimizes static Spectral Risk Measures (SRM). SRM, expressed as a convex combination of CVaRs at varying risk levels, provides practitioners with the flexibility to define a wide range of risk profiles, including the well-known Mean-CVaR measure.

- We demonstrate that return distributions in the DRL framework enable the temporal decomposition of SRM, allowing us to identify intermediate risk measures that preserve the optimality of the policy. These risk measures provide insights into the agent's evolving risk preferences over time and enhance the interpretability of our algorithm.

- Through extensive evaluations, we show that our model accurately learns policies aligned with the SRM objective and outperforms both risk-neutral and risk-sensitive DRL models in various settings.

## 2. Related Works

In this study, we focus on discovering policies with the highest risk-adjusted value:

$$\max_{\pi \in \boldsymbol{\pi}} \rho(Z^{\pi}). \tag{1}$$

Here, $\rho(Z^{\pi})$ denotes the risk-adjusted value of the return of policy $\pi$ and $\boldsymbol{\pi}$ denotes the set of history-dependent policies. In general, optimal policies may depend on all available information up to the current time step. In cases such as risk-neutral RL, optimal policies are typically stationary and Markovian. However, in risk-sensitive RL, the situation is more complex. For instance, in the static CVaR case,

Bäuerle & Ott (2011) demonstrates that the optimal policy depends on the history through a single statistic. By using the representation of CVaR introduced by Rockafellar & Uryasev (2000), they reduce the problem to an ordinary Markov Decision Process (MDP) with an extended state space:

$$\max_{\pi \in \boldsymbol{\pi}} \text{CVaR}_{\alpha}(Z^{\pi}) = \max_{\pi \in \boldsymbol{\pi}} \max_{b \in \mathbb{R}} \left( b + \frac{1}{\alpha} \mathbb{E}\left[ \left[ Z^{\pi} - b \right]^{-} \right] \right)$$
$$= \max_{b \in \mathbb{R}} \left( b + \frac{1}{\alpha} \max_{\pi \in \boldsymbol{\pi}} \mathbb{E}\left[ \left[ Z^{\pi} - b \right]^{-} \right] \right) \tag{2}$$

where $u(z) : z \mapsto [z - b]^{-}$ denote a utility function that is 0 if $z > b$, and $z - b$ otherwise. For a fixed policy, the supremum is attained at $b = F_{Z^{\pi}}^{-1}(\alpha)$. With this representation, the problem is divided into inner and outer optimization problems, with the inner optimization addressing policy search for a fixed parameter $b$, while the outer optimization seeks the optimal parameter $b$.

Bäuerle & Rieder (2014) and Bäuerle & Glauner (2021) extend the idea of state augmentation to the case with a continuous and strictly increasing utility function and SRM as the risk measures. Their work demonstrates that sufficient statistics for solving these problems are cumulative discounted reward and the discount factor up to the decision time. In each of these studies, state augmentation plays a crucial role in formulating a Bellman equation to solve the inner optimization. Regarding the outer optimization in the CVaR case, only the existence of the optimal parameter $b$ is shown. However, for SRM, which needs estimation of an increasing function for the outer optimization, a piecewise linear approximation is used to allow transforming the problem into a finite-dimensional optimization problem. This is then solved using conventional global optimization methods.

In contrast to Bäuerle & Glauner (2021), we use the DRL framework to derive SRM-optimal policies. This framework not only enables the transformation of the problem into a finite-dimensional optimization problem but also allows us to use the closed-form solution to the outer optimization. As we will discuss in section 5, the DRL framework is also essential for identifying the intermediate risk measures for a time-consistent interpretation of optimal policy. While Bäuerle & Ott (2011) demonstrate the existence of these intermediate risk measures for the simple CVaR case, they are not discussed for SRMs in Bäuerle & Glauner (2021).

In the distributional RL framework, Dabney et al. (2018a) introduce the use of risk measures beyond the expectation for action selection. However, their work does not address the theoretical properties of the resulting risk-sensitive policies. For the static CVaR case, Bellemare et al. (2023) adopt

the formulation in Equation 2 and solve the problem using state augmentation. Lim & Malik (2022) focus on a special setting where an optimal Markov CVaR policy exists in the original MDP without requiring state space augmentation. More recently, Kim et al. (2024) study risk-constrained RL, using static SRMs to define the constraints. In contrast to our approach, which leverages a closed-form solution for the outer optimization, they rely on a computationally intensive gradient-based method.

## 3. Preliminary Studies

### 3.1. Spectral Risk Measures

Let $(\Omega, \mathcal{F}, \mathbb{P})$ represent a probability space and $\mathcal{Z}$ represent the space of $\mathcal{F}$-measurable random variables. The historical information available at different time steps is denoted by a filtration $\mathfrak{F} := (\mathcal{F}_t)_{t \geq 0}$ where $\mathcal{F}_s \subset \mathcal{F}_t \subset \mathcal{F}$ for $0 \leq s < t$. We also use $\rho : \mathcal{Z} \to \mathbb{R}$ to denote a risk measure. In the context of this study, $Z$ and $\rho(Z)$ are interpreted as the return and its risk-adjusted value, respectively. Let $F_Z(z) = \mathbb{P}(Z \leq z), z \in \mathbb{R}$ denote the cumulative distribution function (CDF), and $F_Z^{-1}(u) = \inf\{z \in \mathbb{R} : F_Z(z) \geq u\}, u \in [0, 1]$ denote the quantile function of a random variable $Z$. The SRM, introduced by Acerbi (2002), is defined as

$$\text{SRM}_\phi(Z) = \int_0^1 F_Z^{-1}(u)\phi(u)\mathrm{d}u, \tag{3}$$

where the risk spectrum $\phi : [0, 1] \to \mathbb{R}_+$ is a left continuous and non-increasing function with $\int_0^1 \phi(u)\mathrm{d}u = 1$, and denotes the risk preference of the agent. The $\text{CVaR}_\alpha(Z), \alpha \in (0, 1]$ is a special case of SRM with the risk spectrum $\phi(u) = \frac{1}{\alpha}\mathbb{1}_{[0,\alpha]}(u)$. The SRM can also be defined as a convex combination of CVaRs with different risk levels (Kusuoka, 2001). With probability measure $\mu : [0, 1] \to [0, 1]$[1], the SRM can be written as

$$\text{SRM}_\mu(Z) = \int_0^1 \text{CVaR}_\alpha(Z)\mu(\mathrm{d}\alpha). \tag{4}$$

It is shown that an SRM with a bounded spectrum also has a supremum representation

$$\text{SRM}_\phi(Z) = \sup_{h \in \mathcal{H}} \left\{ \mathbb{E}[h(Z)] + \int_0^1 \hat{h}(\phi(u))\mathrm{d}u \right\} \tag{5}$$

where $\mathcal{H}$ denotes the set of concave functions $h : \mathbb{R} \to \mathbb{R}$ and $\hat{h}$ is the concave conjugate of $h$ (Pichler, 2015). In this formulation, the supremum is attained in $h_{\phi,Z} : \mathbb{R} \to \mathbb{R}$

which satisfies $\int_0^1 \hat{h}_{\phi,Z}(\phi(u))\mathrm{d}u = 0$:[2]

$$h_{\phi,Z}(z) = \int_0^1 F_Z^{-1}(\alpha) + \frac{1}{\alpha}\left(z - F_Z^{-1}(\alpha)\right)^- \mu(\mathrm{d}\alpha). \tag{6}$$

### 3.2. Markov Decision Process

In this work, we aim to solve an infinite horizon discounted MDP problem presented by $(\mathcal{X}, \mathcal{A}, \mathcal{R}, \mathcal{P}, \gamma, x_0)$. In this tuple, $\mathcal{X}$ and $\mathcal{A}$ denote the state and action spaces, $\mathcal{R} : \mathcal{X} \times \mathcal{A} \to \mathscr{P}(\mathbb{R})$ the reward kernel, and $\mathcal{P} : \mathcal{X} \times \mathcal{A} \to \mathscr{P}(\mathcal{X})$ the transition kernel, and $\gamma \in [0, 1)$ the discount factor. Without loss of generality, we assume a single initial state represented by $x_0$. Additionally, we assume that the rewards are bounded on the interval $[R_{\text{MIN}}, R_{\text{MAX}}]$ and $R_{\text{MIN}} \geq 0$.

Let $G^\pi$ denote the sum of discounted rewards when starting at $X_0$ and following policy $\pi$, i.e., $G^\pi = \sum_{t=0}^\infty \gamma^t R_t$. With $G_{\text{MIN}} = R_{\text{MIN}}/(1 - \gamma)$ and $G_{\text{MAX}} = R_{\text{MAX}}/(1 - \gamma)$, it's easy to see that $G^\pi$ takes on values in $[G_{\text{MIN}}, G_{\text{MAX}}]$. In this work, we aim to optimize the risk-adjusted value of the cumulative discounted reward based on the SRM. Since the formulation of SRM given in Equation 5 is more suitable in the context of policy-dependent returns, we write our objective as

$$\max_{\pi \in \boldsymbol{\pi}} \text{SRM}_\phi(G^\pi) = \max_{\pi \in \boldsymbol{\pi}} \max_{h \in \mathcal{H}} J(\pi, h)$$
$$= \max_{h \in \mathcal{H}} \left( \max_{\pi \in \boldsymbol{\pi}} J(\pi, h) \right). \tag{7}$$

where $J(\pi, h) = \mathbb{E}[h(G^\pi)] + \int_0^1 \hat{h}(\phi(u))\mathrm{d}u$.

In the remainder of this paper, $\max_{\pi \in \boldsymbol{\pi}} J(\pi, h)$ is referred to as the inner optimization and finding the $\max_{h \in \mathcal{H}}(\cdot)$ is referred to as the outer optimization. To solve the inner optimization problem, we can reduce the search space from history-dependent policies in the original MDP to Markov policies in an augmented MDP with an extended state space denoted by $\mathbf{X} := \mathcal{X} \times \mathcal{S} \times \mathcal{C}$ where $\mathcal{S} = [G_{\text{MIN}}, G_{\text{MAX}}]$ represent the space of accumulated discounted rewards and $\mathcal{C} = (0, 1]$ represent the space of discount factors up to the decision time (Rieder & Bäuerle, 2011; Bäuerle & Glauner, 2021; Bastani et al., 2022). The Markov policies in this MDP take the form $\pi_h : \mathcal{X} \times \mathcal{S} \times \mathcal{C} \to \mathscr{P}(\mathcal{A})$, where the subscript $h$ denotes the dependence of the policy on function $h$ in the inner optimization problem and the space of Markov policies in this MDP is denoted by $\boldsymbol{\pi}_{\mathbf{M}}$. With $X_0 = x_0$, $S_0 = 0, C_0 = 1$, the transition structure of this MDP is defined by $A_t \sim \pi_h(\cdot \mid X_t, S_t, C_t)$, $R_t \sim \mathcal{R}(X_t, A_t)$, $X_{t+1} \sim \mathcal{P}(X_t, A_t)$, $S_{t+1} = S_t + C_t R_t$, and $C_{t+1} = \gamma C_t$.

### 3.3. Distributional RL

Distributional Reinforcement Learning is a sub-field of RL that aims to estimate the full distribution of the return, as op-

---

[1]For a bounded and differentiable risk spectrum $\phi$, we have $\mathrm{d}\phi(u) = -\frac{1}{u}\mu(\mathrm{d}u)$ and $\phi(\alpha) = \int_\alpha^1 \frac{1}{u}\mu(\mathrm{d}u)$

[2]Proof is available in Appendix A.

posed to solely its expected value. To estimate the distribution of the return, DRL uses a distributional value function, which maps states and actions to probability distributions over returns. With $\eta^\pi(x, a)$ denoting the distribution of $G^\pi(x, a)$, the distributional Bellman operator is defined as

$$(\mathcal{T}^\pi \eta)(x, a) = \mathbb{E}_\pi \left[ (b_{R,\gamma})_\# \eta(X', A') \mid X = x, A = a \right], \tag{8}$$

where $A' \sim \pi(\cdot)$ and $b_{r,\gamma} : z \mapsto r + \gamma z$. The push-forward distribution $(b_{R,\gamma})_\# \eta(X', A')$ is also defined as the distribution of $b_{R,\gamma}(G^\pi(X', A'))$. There are multiple ways to parameterize the return distribution, such as the Categorical (C51 algorithm, Bellemare et al., 2017) or the Quantile (QR-DQN algorithm, Dabney et al., 2018b) representation. Here, we use the quantile representation as it simplifies the calculation of risk-adjusted values. With $\tau_i = i/N, i = 0, \cdots, N$ representing the cumulative probabilities, the quantile representation is given by $\eta_\theta(x, a) = \frac{1}{N} \sum_{i=1}^N \delta_{\theta_i(x,a)}$, where the distribution is supported by $\theta_i(x, a) = F_{G(x,a)}^{-1} (\hat{\tau}_i), \hat{\tau}_i = (\tau_{i-1} + \tau_i)/2, 1 \le i \le N$.

### 3.4. Decomposition of Coherent Risk Measures

The decomposition theorem presented in Pflug & Pichler (2016) provides a valuable tool for identifying conditional risk preferences. This theorem states that a law-invariant and coherent risk measure $\rho$ can be decomposed as $\rho(Z) = \sup_{\tilde{\xi}} \mathbb{E}[\tilde{\xi} \cdot \rho_{\tilde{\xi}} (Z \mid \mathcal{F}_t)]$, where the supremum is among all feasible random variables $\tilde{\xi}$ satisfying $\mathbb{E}[\tilde{\xi}] = 1$. In this theorem, if $\xi^\alpha$ is the optimal dual variable to compute the CVaR at level $\alpha$, i.e. $\mathbb{E}[\xi^\alpha Z] = \text{CVaR}_\alpha(Z)$ and $0 \le \xi^\alpha \le 1/\alpha, \xi_t^\alpha = \mathbb{E}[\xi^\alpha \mid \mathcal{F}_t]$, and $\xi = \int_0^1 \xi_t^\alpha \mu(\mathrm{d}\alpha)$, the conditional risk preference is given by

$$\rho_\xi (Z \mid \mathcal{F}_t) = \int_0^1 \text{CVaR}_{\alpha \xi_t^\alpha} (Z \mid \mathcal{F}_t) \frac{\xi_t^\alpha \mu(\mathrm{d}\alpha)}{\xi}. \tag{9}$$

In section 5, we show how the return-distribution of each state can be used to calculate $\xi_t^\alpha$. This value can be used to calculate the new risk levels ($\alpha \xi_t^\alpha$) and their weights ($\xi_t^\alpha \mu(\mathrm{d}\alpha)/\xi$) in the intermediate risk preferences. Moreover, a thorough discussion on the decomposability of risk measures and the time-consistency concept can be found in Appendix F.

## 4. The Model

In this section, we propose an RL algorithm called Quantile Regression with SRM (QR-SRM) to solve the optimization problem outlined in Equation 7. In our approach, the function $h$ is fixed to update the return-distribution in the inner optimization, and then the return-distribution is fixed to update the function $h$. The intuition behind our approach is as follows: The risk spectrum $\phi$ determines the agent's risk

---

**Algorithm 1** The QR-SRM Algorithm

**Input:** A random initialization of $\pi_0^*$ or $G^{\pi_0^*}$
**for** $l = 1, 2, \cdots$ **do**
   **Step 1:** (The Closed-form Solution in Equation 6)
      $h_l = \arg\max_h J(\pi_{l-1}^*, h)$
   **Step 2:** (The Inner Optimization (Algorithm 2))
      $\pi_l^* = \arg\max_\pi J(\pi, h_l)$
**end for**

---

preference by assigning different significance to quantiles of the return-distribution of the initial state. Fixing the function $h$, as displayed in Equation 6, can be interpreted as fixing the estimation of the return-distribution of the initial state. With this estimation, we can solve the inner optimization and find the optimal policy and its associated distributional value function. Since $G^\pi$ is approximated for each state-action pair, we can extract the quantiles of the initial state return-distribution and leverage the closed-form solution presented in Equation 6 to update our estimation of the function $h$ in the outer optimization.[3] Algorithm 1 presents an overview of our method. Note that an important property of the outer optimization's closed-form solution that we will use throughout this section is that $\int_0^1 \hat{h}_{\phi,G}(\phi(u))\mathrm{d}u = 0$ for any return-distribution $G$.

For the inner optimization, let $\eta \in \mathscr{P}(\mathbb{R})^{\mathcal{X} \times \mathcal{S} \times \mathcal{C} \times \mathcal{A}}$ represent the return-distribution function over the augmented state-action space. We denote the corresponding return variable instantiated from $\eta$ as $G$. The greedy selection rule, denoted by $\mathcal{G}_h$, highlights its dependence on the function $h$. The greedy action at the augmented state $(x, s, c)$ is then given by:

$$\mathrm{a}_{G,h}(x, s, c) = \arg\max_{a \in \mathcal{A}} \mathbb{E}\left[ h\left( s + cG(x, s, c, a) \right) \right]. \tag{10}$$

Since function $h$ is fixed until the optimal policy associated with it is found, we can analyze the convergence of the inner optimization with the Bellman operator $\mathcal{T}^{\mathcal{G}_h}$ separately and then discuss the convergence of the overall algorithm.

We use the index $k$ and $l$ to show iterations on $\eta$ and $h$, respectively. Therefore, $\eta_{k,l}$ denotes the $k$th iteration of return-distribution approximation when $h_l$ is used for greedy action selection and $\mathcal{T}^{\mathcal{G}_l}$ denotes the distributional Bellman operator associated with $h_l$. This way, the algorithm begins by setting $\eta_{0,0}(x, s, c, a) = \delta_0$ for all $x \in \mathcal{X}, s \in \mathcal{S}, c \in \mathcal{C}$, and $a \in \mathcal{A}$, initializing $h_0$ based on Equation 6, and iterating $\eta_{k+1,l} = \mathcal{T}^{\mathcal{G}_l} \eta_{k,l}$. This iteration can also be expressed in

---

[3]This is in contrast with the work of Bäuerle & Glauner (2021), which relies on global optimization methods for the outer optimization. Additionally, estimating $G^\pi$ for each state is crucial for identifying intermediate risk measures that ensure a time-consistent interpretation of the optimal policy.

terms of random-variable functions

$$G_{k+1,l}(x,s,c,a) \overset{\mathcal{D}}{=} R(x,a) + \gamma G_{k,l}\left(X',S',C',\mathrm{a}_{k,l}\left(X',S',C'\right)\right), \quad (11)$$

where $\mathcal{D}$ shows equality in distribution and $\mathrm{a}_{k,l}$ denotes the action selection with $G_{k,l}$ and $h_l$.

For the outer optimization, if the optimal policy derived with fixed function $h_l$ is denoted by $\pi_l^*$ and the return-variable of this policy is denoted by $G^{\pi_l^*}$, the iteration on function $h$ is given by

$$h_{l+1} = \arg\max_{h \in \mathcal{H}} J(\pi_l^*, h) \quad (12)$$

Since the supremum in this optimization takes the form of Equation 6, this iteration can be viewed as updating function $h$ with the return distribution of the initial state-action with the highest $\mathrm{SRM}(G^{\pi_l^*})$. The following theorem discusses the convergence of our approach and its proof is provided in Appendix B and C.

**Theorem 4.1.** *If $\pi_{k,l}$ denotes the greedy policy extracted from $G_{k,l}$ and $h_l$, then for all $x \in \mathcal{X}, s \in \mathcal{S}, c \in \mathcal{C}$, and $a \in \mathcal{A}$,*

$$J(\pi_{k,l}, h_l) \geq \max_{\pi \in \boldsymbol{\pi}_{\mathbf{M}}} J(\pi, h_l) - \phi(0)c\gamma^{k+1}G_{\mathrm{MAX}} \quad (13)$$

*Additionally, $J(\pi_l^*, h_l)$ is bounded and monotonically increases as $l$ increases and provides a lower bound for our objective.*

It is important to highlight the distinction between our convergence results and those established for the CVaR case in Bellemare et al. (2023). A key property of CVaR that enables convergence to the optimal policy is that the information required to find the optimal solution can be summarized in a single variable. In the CVaR case, let $q_\alpha$ denote the $\alpha$-quantile in function $h$. Under this formulation, the greedy action selection in Equation 10 can be simplified to the following equation, which aligns with the CVaR greedy policy discussed in Bellemare et al. (2023):

$$\mathrm{a}_{G,h}(x,s,c) = \arg\max_{a \in \mathcal{A}} \mathbb{E}\left[\left(G(x,s,c,a) - \frac{q_\alpha - s}{c}\right)^-\right].$$

Note that in this case, we only need to track a single variable, $\frac{q_\alpha - s}{c}$, rather than all three variables $q_\alpha$, $s$, and $c$, which simplifies action selection. In Bellemare et al. (2023), this variable is denoted by $b$, as seen in Equation 2. The significance of using a single variable becomes evident in their work (Bellemare et al., 2023, Lemma 7.26), where finding the initial $b$ requires searching over all possible values of $b$. This step is crucial for proving convergence to the optimal solution for CVaR.

From a theoretical perspective, for SRM, if we extend the state space to include every quantile required to define $h$,

**Algorithm 2** The Sample Loss For The Inner Optimization of QR-SRM

---

**Input:** $\gamma, \tilde{\theta}, \tilde{\mu}, \theta, (x, s, c, a, r, x')$
$s' \leftarrow s + cr$
$c' \leftarrow \gamma c$
$Q(x', s', c', a') := \frac{1}{N}\sum_{i,j}\tilde{\mu}_i\left(s' + c'\theta_j(x', s', c', a') - \tilde{\theta}_i\right)^-$
$a^* \leftarrow \arg\max_{a'} Q(x', s', c', a')$
$\mathcal{T}^{\mathcal{G}_l}\theta_j(x, s, c, a) \leftarrow r + \gamma\theta_j(x', s', c', a^*), j = 1 \ldots N$
**Output:** $\sum_{i=1}^{N}\mathbb{E}_j\left[\rho_{\hat{\tau}_i}^\kappa\left(\mathcal{T}^{\mathcal{G}_l}\theta_j(x, s, c, a) - \theta_i(x, s, c, a)\right)\right]$

---

we can perform a similar search. However, this approach is computationally expensive and impractical. Therefore, we do not adopt this method in our work and instead focus on a more scalable approach that balances theoretical soundness with practical feasibility.

Algorithm 2 outlines the sample loss for the inner optimization problem. A detailed discussion on the convergence of this Algorithm is available in Appendix E. It is evident that, compared to the sample loss of the risk-neutral QR-DQN algorithm, the only difference lies in the extended state space and action selection. In this algorithm, function $h$ is defined by the return distribution of the initial state $\tilde{\theta}_i := F_{\tilde{G}}^{-1}(\hat{\tau}_i)$, where $\tilde{G} := G^{\pi_{l-1}^*}$. Also, $\tilde{\mu}_i := \int_{\tau_{i-1}}^{\tau_i}\frac{1}{\alpha}\mu(\mathrm{d}\alpha) = \phi(\tau_{i-1}) - \phi(\tau_i)$ denotes the significance of each quantile. The derivation of the action selection in this algorithm from function $h$ is presented in Appendix D.

## 5. Intermediate Risk Preferences

In this section, we discuss the behavior of the optimal policy by identifying the intermediate risk measures for which the policy is optimized. We note that the calculations discussed here do not introduce any computational overhead in the optimization process and are provided solely to enhance the interpretability of our model. Suppose that $G$ and $G_t$ represent $G^{\pi^*}(x_0, 0, 1)$ and $G^{\pi^*}(x_t, s_t, c_t)$, respectively, with $\pi^*$ denoting the optimal policy. In the context of static SRM, the agent's risk preference is defined by assigning weights to the quantiles of $G$. To compute the weights for the quantiles of $G_t$, we establish the relationship between these two return variables, which is where state augmentation becomes crucial.

Suppose the partial random return is denoted by $G_{k:k'} = \sum_{t=k}^{k'}\gamma^{t-k}R_t$ for $k \leq k'$ and $k, k' \in \mathbb{N}$. In traditional RL, the random return is decomposed into the one-step reward and the rewards obtained later: $G_{0:\infty} = R_0 + \gamma G_{1:\infty}$. With $G^\pi(x) \overset{\mathcal{D}}{=} G_{0:\infty}$, the Markov property of the MDP allows writing this decomposition as $G^\pi(x) \overset{\mathcal{D}}{=} R_0 + \gamma G^\pi(X_1), X_0 = x$. In the extended MDP, $\pi \in \boldsymbol{\pi}_{\mathbf{M}}$ also has the Markov property, therefore we have the flexibility to

break down the overall return into the $t$-step reward and the rewards acquired after time $t$, $G_{0:\infty} = G_{0:t-1} + \gamma^t G_{t:\infty}$, and write

$$G^\pi(x_0, 0, 1) \overset{\mathcal{D}}{=} S_t + C_t G^\pi(X_t, S_t, C_t). \quad (14)$$

Since $G$ represents the average of $s_t + c_t G_t$ across all states, for any state $(x_t, s_t, c_t)$ and any quantile level $\alpha$, we can determine the quantile level $\beta$ for $G_t$ such that $F_G^{-1}(\alpha) = s_t + c_t F_{G_t}^{-1}(\beta)$. The following theorem shows that $\xi_t^\alpha$ is, in fact, the ratio of these quantile levels ($\beta/\alpha$), allowing us to define the agent's risk preference at future time steps with respect to $G_t$. The proof of this theorem can be found in Appendix G.

**Theorem 5.1.** *For any SRM defined with probability measure $\mu$, if $\xi^\alpha$ is the optimal dual variable to compute the CVaR at level $\alpha$, i.e. $\mathbb{E}[\xi^\alpha G] = \text{CVaR}_\alpha(G)$, $\lambda_\alpha = F_G^{-1}(\alpha)$ and $F_{G_t}$ is the CDF of $G_t$, we can calculate $\xi_t^\alpha = \mathbb{E}[\xi^\alpha \mid \mathcal{F}_t]$ with:*

$$\xi_t^\alpha = F_{G_t}(\frac{\lambda_\alpha - s_t}{c_t})/\alpha \quad (15)$$

*and derive the risk level and the weight of CVaRs, at a later time step with $\alpha \xi_t^\alpha$ and $\xi_t^\alpha \mu(d\alpha)/\xi$. For the general distributions with discontinuities, calculating $\xi_t^\alpha$ requires an additional term:*

$$\xi_t^\alpha = F_{G_t}(\frac{\lambda_\alpha - s_t}{c_t})/\alpha - \bar{p} \cdot (F_G(\lambda_\alpha) - \alpha)/\alpha \quad (16)$$

*where $\bar{p} = p_{G_t}(\frac{\lambda_\alpha - s_t}{c_t})/p_G(\lambda_\alpha)$.*

Note that with the Quantile representation of return distributions, we have $F_G(\lambda_\alpha) - \alpha \leq 1/N$, therefore the error of omitting the additional term in Equation 16 becomes negligible as the number of quantiles increases. The intuition behind Theorem 5.1 and especially Equation 15 is to find $\xi_t$ for the return distribution of future states ($G_t$) and use the Decomposition Theorem in Section 3.4 to convert this value into the risk levels and weights of CVaRs and its associated risk measure at future states. To further elaborate on the components of the Decomposition theorem and Theorem 5.1, we discuss an example in detail later in this section. We also provide two additional examples in Appendix H, where we use Theorem 5.1 to demonstrate the change in the risk preferences and also analyze a single trajectory in a more practical context within one of our experiments. These examples give a clear intuition behind the temporal adaptation of the risk measure.

**Example 1.** To illustrate the calculations of the conditional risk measures, consider the Markov process in Figure 1, where the number on the edges and nodes represent the transition probabilities and rewards, respectively. In this example, we use $\gamma = 0.5$. For instance, the trajectory

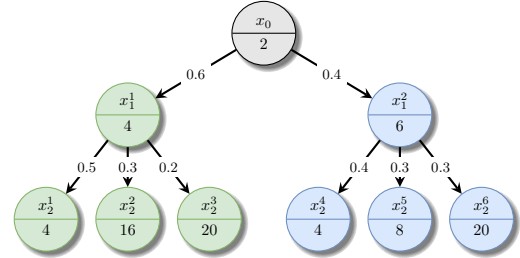

Figure 1: A Markov process with the transition probabilities and rewards denoted on the edges and nodes. This process can also be considered as an MDP with a deterministic policy $\pi$. In this way, the number in each node denotes the $r(x, \pi(x))$.

$(x_0, x_1^1, x_2^3)$ has the reward of $9 = 2 + 0.5 \cdot 4 + 0.5^2 \cdot 20$ and the probability of $0.6 \cdot 0.2 = 0.12$.

Suppose the risk measure has the following form:

$$\rho(G) = 0.7 \cdot \text{CVaR}_{0.4}(G) + 0.3 \cdot \text{CVaR}_{0.8}(G)$$

Here, a direct calculation of the risk measure shows that $\text{CVaR}_{0.4}(G) = 5.25$ and $\text{CVaR}_{0.8}(G) = 6.375$, therefore we have

$$\rho(G) = 0.7 \cdot 5.25 + 0.3 \cdot 6.375 = 5.5875.$$

We now apply the Decomposition theorem to compute this value using conditional risk measures and clarify the notation used in this theorem. The first step involves analyzing how the risk levels $\alpha$ and the weights of each $\text{CVaR}_\alpha$ evolve at $t = 1$. Table 1 presents these calculations, where $\xi^\alpha$ is the optimal dual variable to compute the CVaR at level $\alpha$ satisfying $\mathbb{E}[\tilde{\xi}] = 1$ and $0 \leq \xi^\alpha \leq 1/\alpha$, i.e. $\mathbb{E}[\xi^\alpha Z] = \text{CVaR}_\alpha(Z)$. Additionally, we have $\xi_t^\alpha = \mathbb{E}[\xi^\alpha \mid \mathcal{F}_t]$ and $\xi = \int_0^1 \xi_t^\alpha \mu(d\alpha) = 0.7 \cdot \xi_t^{0.4} + 0.3 \cdot \xi_t^{0.8}$, and the updated risk levels and weights are computed using $\alpha \xi_t^\alpha$ and $\xi_t^\alpha \mu(d\alpha)/\xi$.

Table 1: Details used to compute the conditional risk measures.

| $p_G(z)$ | $G$ | $\xi^{0.4}$ | $\xi^{0.8}$ | $X_1$ | $p_{G_t}(z)$ | $G_t$ | $\xi_t^{0.4}$ | $\xi_t^{0.8}$ | $\xi$ |
|---|---|---|---|---|---|---|---|---|---|
| 30% | 5 | 2.5 | 1.25 | | 50% | 6 | | | |
| 18% | 8 | 0 | 1.25 | $x_1^1$ | 30% | 12 | 1.25 | 1.0833 | 1.2 |
| 12% | 9 | 0 | 0.41667 | | 20% | 14 | | | |
| 16% | 6 | 1.5625 | 1.25 | | 40% | 8 | | | |
| 12% | 7 | 0 | 1.25 | $x_1^2$ | 30% | 10 | 0.625 | 0.875 | 0.7 |
| 12% | 10 | 0 | 0 | | 30% | 16 | | | |

For instance, in state $x_1^1$, we have $\xi_t^{0.4} = 1.25$ and $\xi_t^{0.8} = 1.0833$. With $\xi = 0.7 \cdot 1.25 + 0.3 \cdot 1.0833 = 1.2$, the conditional risk measure at this state is calculated as

$\rho_\xi(G_t \mid X_0 = x_1^1)$

$$= \frac{1.25}{1.2} \cdot \text{CVaR}_{0.5}(G_t) + \frac{1.0833}{1.2} \cdot \text{CVaR}_{0.86}(G_t)$$

$$= 0.73 \cdot \text{CVaR}_{0.5}(G_t) + 0.27 \cdot \text{CVaR}_{0.86}(G_t)$$

$$= 0.73 \cdot 6 + 0.27 \cdot 8.69 = 6.73$$

Similarly for $x_1^2$, we have

$$\rho_\xi(G_t \mid X_0 = x_1^2)$$
$$= 0.625 \cdot \text{CVaR}_{0.25}(G_t) + 0.375 \cdot \text{CVaR}_{0.7}(G_t)$$
$$= 0.625 \cdot 8 + 0.375 \cdot 8.86 = 8.32.$$

Recall that $\rho(G) = \mathbb{E}[\xi \cdot \rho_\xi(G \mid \mathcal{F}_t)]$. Thus, to compute $\rho(G)$, the probabilities of reaching $x_1^1$ or $x_1^2$ are reweighted using $\xi$. Moreover, since the return distribution of the initial state, conditioned on being at $t = 1$, is given by $s_t + c_t G_t$, with $s_t = 2$ and $c_t = 0.5$, we obtain the same value for $\rho(G)$:

$$\rho(G) = 2 + 0.5 \cdot (0.6 \cdot 1.2 \cdot 6.73 + 0.4 \cdot 0.7 \cdot 8.32) = 5.5875$$

Now, consider the goal of analyzing the evolution of the risk measure over time. With this goal in mind, we only need to calculate the updated risk levels and weights using $\xi_t^\alpha$. Theorem 5.1 demonstrates that, rather than directly computing $\xi_t^\alpha$ using $\mathbb{E}[\xi^\alpha \mid \mathcal{F}_t]$, we can leverage the CDF of return variables, as outlined in Equations 15 and 16, to perform the calculations. For $X_1 = x_1^1$, we have:

$$\xi_t^{0.4} = F_{G_t}(\frac{6-2}{0.5})/0.4 = F_{G_t}(8)/0.4 = 0.5/0.4 = 1.25,$$

$$\xi_t^{0.8} = F_{G_t}(\frac{9-2}{0.5})/0.8 - \frac{p_{G_t}(\frac{9-2}{0.5})}{p_G(9)}(F_G(9) - 0.8)/0.8$$
$$= 1/0.8 - \frac{0.2}{0.12}(0.88 - 0.8)/0.8 = 1.0833,$$

and for $X_1 = x_1^2$, we have:

$$\xi_t^{0.4} = F_{G_t}(\frac{6-2}{0.5})/0.4 - \frac{p_{G_t}(\frac{6-2}{0.5})}{0.16}(F_G(6) - 0.4)/0.4$$
$$= 0.4/0.4 - \frac{0.4}{0.16}(0.46 - 0.4)/0.4 = 0.25/0.4 = 0.625,$$
$$\xi_t^{0.8} = F_{G_t}(\frac{9-2}{0.5})/0.8 = F_{G_t}(14)/0.8 = 0.7/0.8 = 0.875.$$

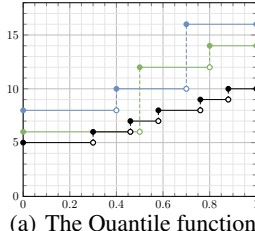
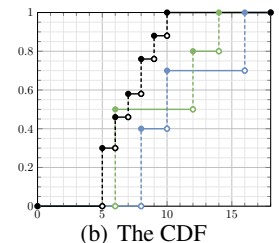

(a) The Quantile function      (b) The CDF

Figure 2: The Quantile function and the CDF of the return-distributions in states $x_0$ (black), $x_1^1$ (green), and $x_1^2$ (blue) in Example 1.

# 6. Experimental Results

In this section, we study our model's performance with four examples. First, we start with the American Option Trading environment, commonly used in the RSRL literature (Tamar et al., 2017; Chow & Ghavamzadeh, 2014; Lim & Malik, 2022), followed by the Mean-reversion trading environment as outlined in the work by Coache & Jaimungal (2023). Finally, we tackle the more challenging Windy Lunar Lander environment. Ultimately, in Appendix I, we also examine the effect of the number of quantiles on performance. The details of these environments are available in Appendix J.

For each experiment, we report the expected value and the risk-adjusted value of the discounted return. Additionally, we employ a diverse range of risk spectrums to derive policies in our algorithm. We denote this approach as QR-SRM($\phi$), where $\phi$ represents the risk spectrum, with subscripts indicating the functional form. The specific cases are as follows:

- QR-SRM($\phi_\alpha$): CVaR with $\phi_\alpha(u) = \frac{1}{\alpha}\mathbb{1}_{[0,\alpha]}(u)$,
- QR-SRM($\phi_{\vec{\alpha},\vec{w}}$): Weighted Sum of CVaRs (WSCVaR) with $\phi_{\vec{\alpha},\vec{w}}(u) = \sum_i w_i \frac{1}{\alpha_i}\mathbb{1}_{[0,\alpha_i]}(u)$,
- QR-SRM($\phi_\lambda$): Exponential risk measure (ERM) with $\phi_\lambda(u) = \frac{\lambda e^{-\lambda u}}{1 - e^{-\lambda}}$,
- QR-SRM($\phi_\nu$): Dual Power risk measure (DPRM) with $\phi_\nu(u) = \nu(1-u)^{\nu-1}$.

## 6.1. American Put Option Trading

In this environment, we assume that the price of the underlying asset follows a Geometric Brownian Motion and at each time step, the the option-holder can either exercise or hold the option. For this example, we selected QR-SRM($\phi_\alpha$) with $\alpha \in \{0.2, 0.6, 1.0\}$. The distribution of option payoff is displayed in Figure 3(a). In this figure, the solid, dashed, and dotted vertical lines depict the $\text{CVaR}_{1.0}(G)$, $\text{CVaR}_{0.6}(G)$, and $\text{CVaR}_{0.2}(G)$ for each of these distributions, respectively. We can see that QR-SRM($\phi_{\alpha=1.0}$), QR-SRM($\phi_{\alpha=0.6}$), and QR-SRM($\phi_{\alpha=0.2}$) successfully finds the policy with the highest $\text{CVaR}_{1.0}(G)$, $\text{CVaR}_{0.6}(G)$ and $\text{CVaR}_{0.2}(G)$, respectively. The exercise boundary of each policy, as depicted in Figure 3(b), also shows that as $\alpha$ decreases from 1.0 to 0.6 and then to 0.2, the policy becomes more conservative and the agent exercises the option sooner, leading to higher $\text{CVaR}_{0.2}(G)$ but lower $\text{CVaR}_{1.0}(G)$.

## 6.2. Mean-reversion Trading Strategy

In the algorithmic trading framework, the asset price follows a mean-reverting process and the agent can buy or sell the asset to earn reward. To showcase our model's performance and its versatility to employ a diverse range of risk spectrums for policy derivation, we employ QR-SRM($\phi_{\lambda=12.0}$), QR-SRM($\phi_{\nu=4.0}$), and QR-SRM($\phi_{\vec{\alpha}_2,\vec{w}_2}$)

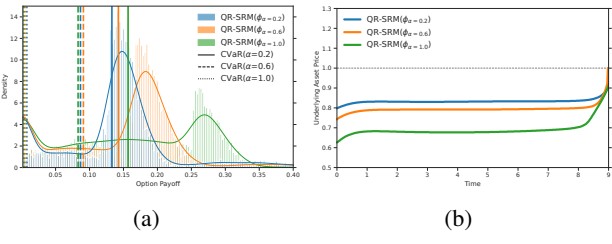

(a)        (b)

Figure 3: Figure 3(a) illustrates the distribution of discounted returns for different policies. Figure 3(b) demonstrates the exercise boundary of each policy.

with $\vec{\alpha}_2$=[0.1, 0.6, 1.0] and $\vec{w}_2$=[0.2, 0.3, 0.5]. In Figure 4(a), the solid, dashed, and dotted vertical lines depict the $\mathrm{WSCVaR}_{\vec{\alpha}_2}^{\vec{w}_2}(G)$, $\mathrm{ERM}_{12.0}(G)$, and $\mathrm{DPRM}_{4.0}(G)$ for each of these distributions, respectively. This figure demonstrates that our model effectively handles more complex risk measures and identifies the policy with the highest $\mathrm{SRM}(G)$.

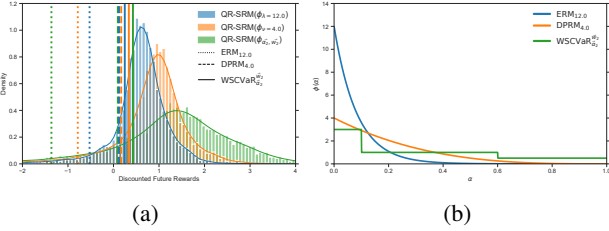

(a)        (b)

Figure 4: Figure 4(a) illustrates the distribution of discounted returns for different policies. Figure 4(b) displays the risk spectrums used to derive these policies.

For this example, we also conduct a comparative evaluation of our model against the risk-neutral QR-DQN model, its risk-sensitive variant introduced in Bellemare et al. (2023) for static CVaR, and a model with risk-sensitive action selection similar to the approach proposed by Dabney et al. (2018a). We refer to these two models as QR-CVaR and QR-iCVaR. As the QR-iCVaR with $\alpha$=1 is identical to the QR-DQN model, only the results of one of them are displayed.

The first three columns of Table 2 represent the risk-adjusted values w.r.t $\mathrm{CVaR}_\alpha$ metric with $\alpha \in \{1.0, 0.5, 0.2\}$. As expected, QR-SRM with CVaR as the risk measure and QR-CVaR exhibit similar performances. On the contrary, QR-iCVaR shows sub-optimal results for $\alpha$=0.5. Even for $\alpha$ values of 0.2 and 1.0, where the average risk-adjusted values of all three models are close, QR-iCVaR achieves lower risk-adjusted value w.r.t other risk measures. Also, a comparison between QR-SRM($\phi_{\alpha=1.0}$) and QR-DQN shows that our model discovers superior policies w.r.t various risk

measures.

For $\mathrm{ERM}_{4.0}$, $\mathrm{DPRM}_{2.0}$, and $\mathrm{WSCVaR}_{\vec{\alpha}_2}^{\vec{w}_2}$, our model can identify top-performing policies. Furthermore, we train a QR-SRM($\phi_{\vec{\alpha}_3, \vec{w}_3}$) algorithm with $\vec{\alpha}_3$=[0.2, 1.0] and $\vec{w}_3$=[0.5, 0.5]. Compared to QR-SRM($\phi_{\alpha=1.0}$) and QR-SRM($\phi_{\alpha=0.2}$) models, the results of this policy demonstrate the possibility of increasing the performance w.r.t to one risk measure at the expense of decreasing the performance w.r.t to another in our model.

### 6.3. Windy Lunar Lander

To evaluate our algorithm in a more complex environment, we utilize the windy Lunar Lander environment. The combination of the larger state and action spaces, along with the stochasticity in the transitions, makes this environment particularly challenging for training. As shown in Table 3, QR-SRM($\phi_{\alpha=1.0}$) performs slightly worse than QR-DQN, but the difference is within a standard deviation. This is likely due to the state augmentation used in our model. We also observed unusually low scores for the QR-CVaR algorithm, which were traced back to poor performance in 3 out of 5 seeds. Additionally, QR-iCVaR under-performed compared to QR-SRM at the same risk levels, suggesting that using a fixed risk measure, as in Dabney et al. (2018b), can lead to sub-optimal performance.

Several factors can contribute to these discrepancies between the objective and the evaluated performance. These include the use of function approximation for value functions and the inherent stochasticity of the environment. However, a particularly significant factor for CVaR is the fact that these objectives focus exclusively on the left tail of the distribution, overlooking valuable information in the right tail. As a result, these algorithms are more likely to converge to sub-optimal policies. This limitation, commonly referred to as "Blindness to Success" (Greenberg et al., 2022), is a well-known issue with CVaR-based approaches.

A remedy to this situation, enabled by our model, is assigning a weight to the expected value. The results for the QR-SRM($\phi_{\vec{\alpha}_3, \vec{w}_3}$) model show that this agent can not only achieve the highest $\mathrm{WSCVaR}_{\vec{\alpha}_3}^{\vec{w}_3}$ but also improve the $\mathrm{CVaR}_{0.2}$ and $\mathrm{CVaR}_{0.5}$ without a great impact on the expected return. The results of this experiment show the impact of having a model with a flexible objective that can adapt to the environment.

## 7. Conclusion

In this paper, we introduced a novel DRL algorithm with convergence guarantees designed to optimize the static SRM. Our empirical evaluations demonstrate the algorithm's ability to learn policies aligned with SRM objectives, achieving superior performance compared to existing methods in a

Table 2: The performance of our model against the QR-DQN, QR-CVaR, and QR-iCVaR models. Bold numbers represent the highest average score with respect to a risk measure. The $\pm$ symbol indicates the standard deviation across seeds.

| Model | $\text{CVaR}_{1.0}$ | $\text{CVaR}_{0.5}$ | $\text{CVaR}_{0.2}$ | $\text{ERM}_{4.0}$ | $\text{DPRM}_{2.0}$ | $\text{WSCVaR}^{\vec{w}_2}_{\vec{\alpha}_2}$ | $\text{WSCVaR}^{\vec{w}_3}_{\vec{\alpha}_3}$ |
|---|---|---|---|---|---|---|---|
| QR-SRM($\phi_{\alpha=1.0}$) | 1.43±0.03 | 0.03±0.04 | -1.36±0.09 | -0.37±0.05 | 0.43±0.02 | 0.35±0.02 | 0.04±0.04 |
| QR-CVaR($\alpha$=1.0) | **1.48±0.07** | -0.02±0.10 | -1.42±0.21 | -0.41±0.13 | 0.40±0.07 | 0.35±0.08 | 0.03±0.10 |
| QR-DQN | 1.40±0.09 | -0.24±0.17 | -1.76±0.27 | -0.67±0.19 | 0.21±0.14 | 0.16±0.14 | -0.18±0.16 |
| QR-SRM($\phi_{\alpha=0.5}$) | 0.78±0.02 | 0.27±0.03 | -0.44±0.05 | 0.00±0.04 | 0.38±0.02 | 0.29±0.03 | 0.17±0.03 |
| QR-CVaR($\alpha$=0.5) | 0.79±0.08 | **0.28±0.02** | -0.41±0.11 | 0.02±0.05 | 0.40±0.03 | 0.31±0.02 | 0.19±0.03 |
| QR-iCVaR($\alpha$=0.5) | 0.82±0.17 | 0.14±0.04 | -0.36±0.09 | -0.00±0.02 | 0.32±0.07 | 0.33±0.07 | 0.23±0.05 |
| QR-SRM($\phi_{\alpha=0.2}$) | 0.56±0.08 | 0.21±0.03 | -0.21±0.04 | **0.05±0.01** | 0.29±0.05 | 0.24±0.04 | 0.17±0.02 |
| QR-CVaR($\alpha$=0.2) | 0.64±0.06 | 0.24±0.03 | -0.27±0.03 | **0.05±0.01** | 0.33±0.03 | 0.26±0.03 | 0.19±0.02 |
| QR-iCVaR($\alpha$=0.2) | 0.40±0.04 | 0.04±0.01 | **-0.17±0.03** | -0.00±0.01 | 0.13±0.02 | 0.16±0.02 | 0.11±0.02 |
| QR-SRM($\phi_{\lambda=4.0}$) | 0.84±0.05 | **0.28±0.02** | -0.37±0.07 | **0.05±0.03** | 0.42±0.02 | 0.35±0.02 | 0.23±0.03 |
| QR-SRM($\phi_{\nu=2.0}$) | 1.01±0.09 | 0.27±0.02 | -0.62±0.14 | -0.03±0.06 | 0.46±0.02 | 0.36±0.01 | 0.19±0.03 |
| QR-SRM($\phi_{\vec{\alpha}_2,\vec{w}_2}$) | 1.06±0.11 | 0.26±0.03 | -0.57±0.14 | -0.01±0.05 | 0.47±0.03 | **0.40±0.02** | **0.24±0.02** |
| QR-SRM($\phi_{\vec{\alpha}_3,\vec{w}_3}$) | 1.11±0.08 | 0.26±0.04 | -0.68±0.19 | -0.05±0.08 | **0.48±0.02** | **0.40±0.03** | 0.22±0.06 |

Table 3: The performance of our model against the QR-DQN, QR-CVaR, and QR-iCVaR models. Bold numbers represent the highest average score with respect to a risk measure. The $\pm$ symbol indicates the standard deviation across seeds.

| Model | $\mathbb{E}$ | $\text{CVaR}_{0.5}$ | $\text{CVaR}_{0.2}$ | $\text{WSCVaR}^{\vec{w}_3}_{\vec{\alpha}_3}$ |
|---|---|---|---|---|
| QR-SRM($\phi_{\alpha=1.0}$) | 27.36±12.85 | 10.17±14.07 | -8.07±17.36 | 9.82±14.75 |
| QR-CVaR($\alpha$=1.0) | 14.52±1.91 | -1.51±3.73 | -15.58±5.73 | -0.45±3.24 |
| QR-DQN | **32.73±8.98** | 3.79±13.72 | -24.79±23.39 | 4.23±15.99 |
| QR-SRM($\phi_{\alpha=0.5}$) | 23.25±12.58 | 4.12±12.64 | -12.26±11.67 | 5.67±12.05 |
| QR-CVaR($\alpha$=0.5) | 10.77±5.34 | -6.26±8.44 | -21.07±13.08 | -5.08±9.13 |
| QR-iCVaR($\alpha$=0.5) | 21.72±24.09 | -1.33±29.33 | -22.37±38.35 | -0.02±30.97 |
| QR-SRM($\phi_{\alpha=0.2}$) | 22.28±12.66 | 3.37±13.09 | -13.51±14.54 | 4.57±13.53 |
| QR-CVaR($\alpha$=0.2) | 11.85±6.73 | -4.78±6.16 | -19.09±4.19 | -3.53±5.32 |
| QR-iCVaR($\alpha$=0.2) | 18.53±22.07 | -2.69±27.34 | -18.82±31.68 | 0.04±26.80 |
| QR-SRM($\phi_{\vec{\alpha}_3,\vec{w}_3}$) | 31.70±11.21 | **14.17±13.46** | **-2.85±20.63** | **14.57±15.22** |

variety of risk-sensitive scenarios. The advantage of using static SRM extends beyond performance; it also enhances interpretability. We showed that by applying the Decomposition Theorem of coherent risk measures and leveraging the return distribution available in the DRL framework, we can identify the specific objective that the optimal policy is optimizing for. This allows for monitoring the policy's behavior and risk sensitivity, and adjusting it if necessary.

A few limitations of our work that can pave the way for future research are as follows: i) Our value-based method is suitable for environments with discrete action spaces. The extension of our algorithm to actor-critic methods can make our approach available in environments with continuous action spaces. ii) In this work, we parameterized the return distribution with the quantile representation. Using other parametric approximations of the distribution (Dabney et al., 2018a; Yang et al., 2019) or improvements that have been introduced for the quantile representation (Zhou et al., 2020; 2021) can potentially improve the performance of our risk-sensitive algorithm. iii) The algorithm to update the function $h$, or equivalently the estimation of the initial state's return distribution, provides a lower bound for

the objective. In section 6.2, we empirically observed that our algorithm converges to policies similar to QR-CVaR, which has stronger convergence guarantees. However, an algorithm with stronger guarantees for convergence to the optimal function $h$ can enhance our understanding of static SRM.

## Impact Statement

This work advances Machine Learning by providing a principled approach to handling worst-case scenarios in sequential decision-making. We do not foresee any negative societal impacts. On the contrary, our method can enhance the safety and reliability of AI systems in high-stakes domains such as finance and healthcare, where effective risk management is critical. We believe this contribution supports the development of safer and more trustworthy machine learning applications.

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

## A. Property of the Closed-form solution

Using the SRM definition from Equation 4, we have

$$
\begin{aligned}
\mathrm{SRM}_\mu(Z) &= \int_0^1 \mathrm{CVaR}_\alpha(Z)\mu(\mathrm{d}\alpha) \\
&\overset{(a)}{=} \int_0^1 F_Z^{-1}(\alpha) + \frac{1}{\alpha}\mathbb{E}\left[\left(Z - F_Z^{-1}(\alpha)\right)^-\right]\mu(\mathrm{d}\alpha) \\
&\overset{(b)}{=} \mathbb{E}\left[\int_0^1 F_Z^{-1}(\alpha) + \frac{1}{\alpha}\left(Z - F_Z^{-1}(\alpha)\right)^- \mu(\mathrm{d}\alpha)\right] \\
&= \mathbb{E}\left[h_{\phi,Z}(Z)\right]
\end{aligned}
$$

where step $(a)$ utilizes the CVaR representation provided in Rockafellar & Uryasev (2000), and step $(b)$ applies Fubini's Theorem. Next, we note that $h_{\phi,Z}$, as defined in Equation 6, is differentiable almost everywhere, with its derivative given by

$$
\begin{aligned}
h'_{\phi,Z}(z) &= \int_{\left\{\alpha:z\leq F_Z^{-1}(\alpha)\right\}} \frac{1}{\alpha}\mu_\phi(\mathrm{d}\alpha) \\
&= \int_{F_Z(z)}^1 \frac{1}{\alpha}\mu_\phi(\mathrm{d}\alpha) = \phi\left(F_Z(z)\right).
\end{aligned}
$$

Additionally, the infimum in the concave conjugate $\hat{h}_{\phi,Z}(\phi(u)) = \inf_z \left(\phi(u)\cdot z - h_{\phi,Z}(z)\right)$ is achieved at any $z$ where $\phi(u) = h'_{\phi,Z}(z) = \phi\left(F_Z(z)\right)$, which corresponds to $z = F_Z^{-1}(u)$. Therefore, we obtain

$$
\begin{aligned}
\int_0^1 \hat{h}_{\phi,Z}(\phi(u))\mathrm{d}u &= \int_0^1 \phi(u)\cdot F_Z^{-1}(u) - h_{\phi,Z}\left(F_Z^{-1}(u)\right)\mathrm{d}u \\
&= \int_0^1 \phi(u)\cdot F_Z^{-1}(u)\mathrm{d}u - \int_0^1 h_{\phi,Z}\left(F_Z^{-1}(u)\right)\mathrm{d}u \\
&= \mathrm{SRM}_\phi(Z) - \mathbb{E}\left[h_{\phi,Z}(Z)\right] \\
&= 0
\end{aligned}
$$

## B. Proof of Convergence for Inner Optimization

In this section, we aim to demonstrate the convergence of the inner optimization algorithm to the optimal policy associated with a fixed function $h_l$. Before discussing the main theorem, we must introduce several intermediate results about partial returns. Recall that $J(\pi, h_l) = \mathbb{E}\left[h_l\left(G^\pi\right)\right] + \int_0^1 \hat{h}_l(\phi(u))\mathrm{d}u$. Since $\int_0^1 \hat{h}_l(\phi(u))\mathrm{d}u = 0$, we define the mapping $V_{k,l} : \mathcal{X} \times \mathcal{S} \times \mathcal{C} \times \mathcal{A} \to \mathbb{R}$ as follows:

$$
V_{k,l}(x, s, c, a) = \mathbb{E}\left[h_l\left(s + cG_{k,l}(x, s, c, a)\right)\right]. \tag{17}
$$

Similarly for a policy $\pi_l \in \boldsymbol{\pi}_{\mathbf{M}}$, we define

$$
V^{\pi_l}(x, s, c, a) = \mathbb{E}\left[h_l\left(s + cG^{\pi_l}(x, s, c, a)\right)\right]. \tag{18}
$$

The goal is to find an optimal deterministic policy $\pi_l^* \in \boldsymbol{\pi}_{\mathbf{M}}$ in the sense that

$$
V^{\pi_l^*}(x, s, c, a) = \max_{\pi_l \in \boldsymbol{\pi}_{\mathbf{M}}} V^{\pi_l}(x, s, c, a). \tag{19}
$$

With $\mathrm{a}_{k,l}(x, s, c) = \mathrm{a}_{G_k, h_l}(x, s, c)$, we have the following recursive property for $V_{k,l}$ and $V^{\pi_l}$:

**Lemma B.1.** *For each* $(x, s, c, a) \in \mathcal{X} \times \mathcal{S} \times \mathcal{C} \times \mathcal{A}$, *we have*

$$
V_{k+1,l}(x, s, c, a) = \mathbb{E}_{xsca}\left[V_{k,l}\left(X', S', C', \mathrm{a}_{k,l}\left(X', S', C'\right)\right)\right]. \tag{20}
$$

*Additionally, for a policy* $\pi_l \in \boldsymbol{\pi}_{\mathbf{M}}$, *we have*

$$
V^{\pi_l}(x, s, c, a) = \mathbb{E}_{\pi_l, xsca}\left[V^{\pi_l}\left(X', S', C', A'\right)\right] \tag{21}
$$

*Proof.* The proof for both equations follows similar steps, so we present the proof only for $V_{k+1,l}$. Consider a partial trajectory that starts with the sample transition $(X, S, C, A, R, X', S', C', A')$ and continues with $(X_t, S_t, C_t, A_t, R_t)_{t=0}^{k}$ in which $A_0 = A' = a_{k,l}(X', S', C')$ and $A_t \sim \pi_{k-t,l}(\cdot \mid X_t, S_t, C_t), t \geq 1$. Since $S' = S + CR$ and $C' = \gamma C$, we can write[4]

$$
\begin{aligned}
&\mathbb{E}_{xsca}[V_{k,l}(X', S', C', A')] \\
&= \mathbb{E}_{xsca}\left[\mathbb{E}\left[h_l\left(S' + C' G_{k,l}(X', S', C', A')\right)\right]\right] \\
&= \mathbb{E}_{xsca}\left[\mathbb{E}\left[h_l\left(S' + C' \sum_{t=0}^{k}\gamma^t R_t\right) \mid X_0 = X', S_0 = S', C_0 = C', A_0 = A'\right]\right] \\
&= \mathbb{E}_{xsca}\left[\mathbb{E}\left[h_l\left(s + cR + \gamma c \sum_{t=0}^{k}\gamma^t R_t\right) \mid X_0 = X', S_0 = S', C_0 = C', A_0 = A'\right]\right] \\
&= \mathbb{E}_{xsca}\left[\mathbb{E}\left[h_l\left(s + cR + c \sum_{t=1}^{k+1}\gamma^t R_t\right) \mid X_1 = X', S_1 = S', C_1 = C', A_1 = A'\right]\right] \\
&= \mathbb{E}\left[h_l\left(s + c \sum_{t=0}^{k+1}\gamma^t R_t\right) \mid X_0 = x, S_0 = s, C_0 = c, A_0 = a\right] \\
&= V_{k+1,l}(x, s, c, a)
\end{aligned}
$$

$\square$

**Lemma B.2.** *For each $(x, s, c, a) \in \mathcal{X} \times \mathcal{S} \times \mathcal{C} \times \mathcal{A}$ and return-distribution $\eta_{k,l}$ defined by Equation 11, the associated $V_{k,l}(x, s, c, a)$ indicates the value of the optimal policy for partial return, i.e.:*

$$
V_{k,l}(x, s, c, a) = \max_{\pi_l \in \boldsymbol{\pi}_{\mathbf{M}}} \mathbb{E}_{\pi_l, xsca}\left[h_l\left(s + c \sum_{t=0}^{k}\gamma^t R_t\right)\right]. \tag{22}
$$

*Proof.* We establish the validity of this lemma through induction on $k$. The statement holds true for $k = 0$ with $G_{0,l}(x, s, c, a) = 0$. Assuming the statement is true for $V_{k,l}$, we leverage the results of Lemma B.1 and the fact that the policy $\mathcal{G}_l(\eta_{k,l})$ selects the action maximizing $V_{k,l}$ to conclude the validity of the statement for $V_{k+1,l}$. $\square$

**Lemma B.3.** *For each $(x, s, c, a) \in \mathcal{X} \times \mathcal{S} \times \mathcal{C} \times \mathcal{A}$, it holds that*

$$
V^{\pi_l^*}(x, s, c, a) - \varepsilon_k(x, s, c, a) \leq V_{k,l}(x, s, c, a) \leq V^{\pi_l^*}(x, s, c, a) \tag{23}
$$

*where $\lim_{k \to \infty} \varepsilon_k(x, s, c, a) = 0$. It also holds that $V_{k,l}(x, s, c, a) \uparrow V^{\pi_l^*}(x, s, c, a)$.*

*Proof.* Let $(R_t)_{t \geq 0}$ be a sequence of rewards in $[R_{\mathrm{MIN}}, R_{\mathrm{MAX}}]$ for any policy $\pi_l$. Since $h_l$ is a non-decreasing function, we have

$$
h_l\left(s + c \sum_{t=0}^{k}\gamma^t R_t\right) \leq h_l\left(s + c \sum_{t=0}^{\infty}\gamma^t R_t\right). \tag{24}
$$

Also, we can use the $\phi(0)$-Lipschitz property of $h_l$, i.e. $h_l(u_1 + u_2) - h_l(u_1) \leq \phi(0)u_2$, to write

$$
h_l\left(s + c \sum_{t=0}^{\infty}\gamma^t R_t\right) - h_l\left(s + c \sum_{t=0}^{k}\gamma^t R_t\right) \leq \phi(0)c \sum_{t=k+1}^{\infty}\gamma^t R_t \leq \phi(0)c\gamma^{k+1}G_{\mathrm{MAX}} \tag{25}
$$

Combining these results yields the following inequality

$$
V_{\infty}^{\pi_l}(x, s, c, a) - \varepsilon_k(x, s, c, a) \leq V_k^{\pi_l}(x, s, c, a) \leq V_{\infty}^{\pi_l}(x, s, c, a) \tag{26}
$$

---

[4]Note that $C_t$ is a degenerate random variable that only takes the value $\gamma^t$, therefore multiplying a random variable $Z$ by $C_t$ scales each realization by $\gamma^t$.

where $\varepsilon_k(x, s, c, a) = \phi(0)c\gamma^{k+1}G_{\text{MAX}}$. This shows that $\lim_{k \to \infty} \varepsilon_k(x, s, c, a) = 0$. Taking the supremum over all policies and applying Lemma B.2 results in Inequality 23. By setting $G_{0,l}(x, s, c, a) = 0$ and considering the non-negativity of rewards, we ensure that $V_{k,l}(x, s, c, a)$ is increasing with respect to $k$ and therefore we have $V_{k,l}(x, s, c, a) \uparrow V^{\pi_l^*}(x, s, c, a)$.  $\square$

**Theorem B.4.** *With $\pi_{k,l} = \mathcal{G}_l(\eta_{k,l})$, it holds that $\lim_{k \to \infty} V^{\pi_{k,l}} = V^{\pi_l^*}$.*

*Proof.* Given that the function $h_l$ is non-decreasing and considering the definitions of $V_{k,l}(x, s, c, a)$ and $V^{\pi_{k,l}}(x, s, c, a)$, we can write:

$$0 \leq \phi(1) \leq \frac{\mathbb{E}[h_l(s + cG_{k,l}(x, s, c, a)) - h_l(s + cG_{k-1,l}(x, s, c, a))]}{c\mathbb{E}[G_{k,l}(x, s, c, a) - G_{k-1,l}(x, s, c, a)]}, \tag{27}$$

and

$$0 \leq \phi(1) \leq \frac{\mathbb{E}[h_l(s + cG^{\pi_{k,l}}(x, s, c, a)) - h_l(s + cG_{k,l}(x, s, c, a))]}{c\mathbb{E}[G^{\pi_{k,l}}(x, s, c, a) - G_{k,l}(x, s, c, a)]}. \tag{28}$$

Since $V_{k,l}(x, s, c, a)$ is increasing w.r.t $k$, the numerator in Equation 27 is positive and we can conclude that $\mathbb{E}[G_{k,l}(x, s, c, a)] \geq \mathbb{E}[G_{k-1,l}(x, s, c, a)]$. Utilizing Equation 11, we can also infer that

$$\mathbb{E}_{xsca}[\mathbb{E}[G_{k,l}(X', S', C', a_{k,l}(X', S', C'))]] \geq \mathbb{E}_{xsca}[\mathbb{E}[G_{k-1,l}(X', S', C', a_{k-1,l}(X', S', C'))]]. \tag{29}$$

Now in Equation 28, in order to show that $V^{\pi_{k,l}}(x, s, c, a) - V_{k,l}(x, s, c, a) \geq 0$ in every state-action, we need to show that the denominator in this equation is also always positive. With $\epsilon_k(x, s, c, a) := \mathbb{E}[G^{\pi_{k,l}}(x, s, c, a) - G_{k,l}(x, s, c, a)]$, we have

$$\begin{aligned}
\epsilon_k(x, s, c, a) &= \mathbb{E}[G^{\pi_{k,l}}(x, s, c, a) - G_{k,l}(x, s, c, a)] \\
&= \mathbb{E}_{xsca}[\mathbb{E}[R + \gamma G^{\pi_{k,l}}(X', S', C', a_{k,l}(X', S', C')) - R - \gamma G_{k-1,l}(X', S', C', a_{k-1,l}(X', S', C'))]] \\
&= \gamma \mathbb{E}_{xsca}[\mathbb{E}[G^{\pi_{k,l}}(X', S', C', a_{k,l}(X', S', C')) - G_{k-1,l}(X', S', C', a_{k-1,l}(X', S', C'))]] \\
&\overset{(a)}{\geq} \gamma \mathbb{E}_{xsca}[\mathbb{E}[G^{\pi_{k,l}}(X', S', C', a_{k,l}(X', S', C')) - G_{k,l}(X', S', C', a_{k,l}(X', S', C'))]] \\
&= \gamma \mathbb{E}_{xsca}[\epsilon_k(X', S', C', a_{k,l}(X', S', C'))],
\end{aligned} \tag{30}$$

where we use Equation 29 for $(a)$. Given that $\epsilon_k(x, s, c, a)$ is bounded from below, its infimum $\epsilon_k := \inf_{(x,s,c,a)} \epsilon_k(x, s, c, a)$ exists, so we can take infimum from both sides of Equation 30 and replace $\epsilon_k(x, s, c, a)$ with $\epsilon_k$. This leads to

$$\epsilon_k \geq \gamma \epsilon_k \implies \epsilon_k \geq 0, \tag{31}$$

demonstrating that both denominator and numerator in Equation 28 are positive. Therefore, we can prove the theorem using the Squeeze Theorem and Lemma B.3.  $\square$

## C. Lower Bound for the Objective

In Appendix B, we showed that the fixed point of each distributional Bellman operator $\mathcal{T}^{\mathcal{G}_l}$ denoted by $\eta_l^*$ and instantiated as $G^{\pi_l^*}$ can be found and we were able to provide the error bound for each $\pi_{k,l}$. Using the fact that $\int_0^1 \hat{h}_l(\phi(u))du = 0$ for $l \in \mathbb{N}$, the update rule for function $h$ in Equation 12 shows

$$\mathbb{E}\left[h_{l+1}\left(G^{\pi_l^*}(X_0, 0, 1, a_{G^{\pi_l^*}, h_{l+1}}(X_0, 0, 1))\right)\right] \geq \mathbb{E}\left[h_l\left(G^{\pi_l^*}(X_0, 0, 1, a_l^*(X_0, 0, 1))\right)\right].$$

where $a_l^*(x, s, c) = a_{G^{\pi_l^*}, h_l}(x, s, c)$ denotes the optimal action when the same function $h_l$ is used to estimate $G^{\pi_l^*}$ and calculate $\mathbb{E}[h_l(\cdot)]$. Remember that the return-variable of the optimal policy derived with the fixed function $h_l$ and $h_{l+1}$ is denoted by $G^{\pi_l^*}$ and $G^{\pi_{l+1}^*}$. Therefore, we have

$$\begin{aligned}
&\mathbb{E}\left[h_{l+1}\left(G^{\pi_{l+1}^*}(X_0, 0, 1, a_{l+1}^*(X_0, 0, 1))\right)\right] \geq \mathbb{E}\left[h_{l+1}\left(G^{\pi_l^*}(X_0, 0, 1, a_{G^{\pi_l^*}, h_{l+1}}(X_0, 0, 1))\right)\right] \\
&\implies \mathbb{E}\left[h_{l+1}\left(G^{\pi_{l+1}^*}(X_0, 0, 1, a_{l+1}^*(X_0, 0, 1))\right)\right] \geq \mathbb{E}\left[h_l\left(G^{\pi_l^*}(X_0, 0, 1, a_l^*(X_0, 0, 1))\right)\right],
\end{aligned}$$

and relative to our objective, we can write:

$$\text{SRM}_\phi\left(G^{\pi_l^*}\right) = \sup_{h \in \mathcal{H}} \mathbb{E}\left[h\left(G^{\pi_l^*}(X_0, 0, 1, \mathrm{a}_l^*(X_0, 0, 1))\right)\right]$$

$$\geq \mathbb{E}\left[h_l\left(G^{\pi_l^*}(X_0, 0, 1, \mathrm{a}_l^*(X_0, 0, 1))\right)\right]$$

Since the rewards are bounded, both $G^{\pi_l^*}$ and function $h_l$ are bounded. Therefore, the monotonic increase of $J(\pi_l^*, h_l) = V^{\pi_l^*}(X_0, 0, 1, \mathrm{a}_l^*(X_0, 0, 1))$ as $l \to \infty$ provides a lower bound for the objective.

## D. Details of Algorithm 2

The quantile regression loss function used in this algorithm helps estimate the quantiles by penalizing both overestimation and underestimation with weights $\tau$ and $1 - \tau$, respectively. The quantile Huber loss function (Huber, 1992) uses the squared regression loss in an interval $[-\kappa, \kappa]$ to prevent the gradient from becoming constant when $u \to 0^+$:

$$\rho_\tau^\kappa(u) = \left|\tau - \delta_{\{u<0\}}\right| \mathcal{L}_\kappa(u) \tag{32}$$

where the Huber loss $\mathcal{L}_\kappa(u)$ is given by

$$\mathcal{L}_\kappa(u) = \left\{ \begin{array}{ll} \frac{1}{2}u^2, & \text{if } |u| \leq \kappa \\ \kappa\left(|u| - \frac{1}{2}\kappa\right), & \text{otherwise} \end{array} \right. . \tag{33}$$

Furthermore, since function $h_l$ is approximated with the quantile representation of $\tilde{G} := G^{\pi_{l-1}}$ and Equation 6, we need to show how $\mathbb{E}\left[h_l(s_k' + c_k'\theta_j(x_k', s_k', c_k', a'))\right]$ is calculated. With $z_j := s_k' + c_k'\theta_j(x_k', s_k', c_k', a')$ and $\tilde{\theta}_i := F_{\tilde{G}}^{-1}(\hat{\tau}_i)$, we can write

$$h_l(z_j) = \int_0^1 F_{\tilde{G}}^{-1}(\alpha) + \frac{1}{\alpha}\left(z_j - F_{\tilde{G}}^{-1}(\alpha)\right)^- \mu(\mathrm{d}\alpha)$$

$$= \sum_i \left(\int_{\tau_{i-1}}^{\tau_i} F_{\tilde{G}}^{-1}(\alpha) + \frac{1}{\alpha}\left(z_j - F_{\tilde{G}}^{-1}(\alpha)\right)^- \mu(\mathrm{d}\alpha)\right)$$

$$= \sum_i \left(\int_{\tau_{i-1}}^{\tau_i} F_{\tilde{G}}^{-1}(\alpha)\mu(\mathrm{d}\alpha) + \int_{\tau_{i-1}}^{\tau_i} \frac{1}{\alpha}\left(z_j - F_{\tilde{G}}^{-1}(\alpha)\right)^- \mu(\mathrm{d}\alpha)\right)$$

$$\overset{(a)}{=} \sum_i \left(\tilde{\theta}_i \int_{\tau_{i-1}}^{\tau_i} \mu(\mathrm{d}\alpha) + \left(z_j - \tilde{\theta}_i\right)^- \int_{\tau_{i-1}}^{\tau_i} \frac{1}{\alpha}\mu(\mathrm{d}\alpha)\right). \tag{34}$$

In this calculation, the integration interval $[0, 1]$ is divided into $N$ intervals $[\tau_0, \tau_1), [\tau_1, \tau_2), \cdots, [\tau_{N-2}, \tau_{N-1}), [\tau_{N-1}, \tau_N]$. Therefore, the integrals $\int_{\tau_{i-1}}^{\tau_i} \mu(\mathrm{d}\alpha)$ is calculated on $[\tau_{i-1}, \tau_i)$, including the lower limit $\tau_{i-1}$ and excluding the upper limit $\tau_i$. In (a), we used the fact that $\tilde{\theta}_i$ is constant in $[\tau_{i-1}, \tau_i)$. Also, the first term in the summation can be omitted since it is constant for all actions.

In this algorithm, it's also important to highlight the direct relationship between the number of quantiles and the expressiveness of SRM. For example, when the return distribution is approximated with $N$ quantiles, the expectation can be estimated with $\tilde{\mu}_N = \int_{1-1/N}^1 \frac{1}{\alpha}\mu(\mathrm{d}\alpha) = 1$ and $\tilde{\mu}_j = 0$ for $1 \leq j < N$. Similarly, $\text{CVaR}_\alpha$ for $\alpha < 1$ can be approximated by setting $\tilde{\mu}_j = 1/\alpha$ for $j = \lfloor \alpha N \rfloor + 1$ and $\tilde{\mu}_j = 0$ otherwise.

## E. Convergence of Algorithm 2

As described in Section 3.3, we parameterize the return distribution using a quantile representation. Specifically, we employ a quantile projection operator, $\Pi_Q$, to map any return distribution $\eta$ onto its quantile representation with respect to the 1-Wasserstein distance ($\text{w}_1$). Therefore, $\Pi_Q \eta = \hat{\eta} = \frac{1}{N}\sum_{i=1}^N \delta_{\theta_i}$ with $\theta_i = F_\eta^{-1}(\hat{\tau}_i), \hat{\tau}_i = (\tau_{i-1} + \tau_i)/2, 1 \leq i \leq N$ corresponds to the solution of the following minimization problem:

$$\text{minimize } \text{w}_1(\eta, \eta') \text{ subject to } \eta' \in \mathscr{F}_{Q,N}$$

where $\mathscr{F}_{Q,N}$ is the space of quantile representations with $N$ quantiles. Using this definition, Algorithm 2 can be expressed as iteratively updating

$$\hat{\eta}_{k+1,l} = \Pi_Q \mathcal{T}^{\mathcal{G}_l} \hat{\eta}_{k,l}.$$

As previously noted, this process is analogous to the iteration in the QR-DQN algorithm, with two key differences: the incorporation of risk-sensitive greedy action selection and the use of an extended state space. Consequently, we can leverage the steps outlined in Bellemare et al. (2023, Section 7.3) to establish the convergence of $\Pi_Q \mathcal{T}^{\mathcal{G}_l}$.

To begin, we will demonstrate that $\mathcal{T}^{\mathcal{G}_l}$ is a contraction mapping. That is, the sequence of iterates defined by $\eta_{k+1,l} = \mathcal{T}^{\mathcal{G}_l} \eta_{k,l}$ converges to $\eta^{\pi_l^*}$ with respect to the supremum $p$-Wasserstein distance, $\bar{w}_p$, for $p \in [1, \infty]$. Here, we assume the existence of a unique optimal policy $\pi_l^*$.[5] With this assumption, we leverage the fact that the action gap, $\mathrm{GAP}(Q)$—defined as the smallest difference between the highest-valued and second-highest-valued actions across all states for a given Q-function—is strictly positive. By setting $\bar{\varepsilon} = \mathrm{GAP}(V^{\pi_l^*})/2$ and using Lemma B.3, we can see that after $K_{\bar{\varepsilon}} \in \mathbb{N}$ iterations where $K_{\bar{\varepsilon}} := \lfloor \ln(\frac{\bar{\varepsilon}}{\phi(0)G_{\mathrm{MAX}}})/\ln(\gamma) \rfloor$, the greedy action in state $(x, s, c)$ becomes the optimal action $a^*$, and for any $a \neq a^*$, we have:

$$\begin{aligned}
V_{k,l}(x, s, c, a^*) &\geq V^{\pi_l^*}(x, s, c, a^*) - \bar{\varepsilon} \\
&\geq V^{\pi_l^*}(x, s, c, a) + \mathrm{GAP}(V^{\pi_l^*}) - \bar{\varepsilon} \\
&> V_{k,l}(x, s, c, a) + \mathrm{GAP}(V^{\pi_l^*}) - 2\bar{\varepsilon} \\
&= V_{k,l}(x, s, c, a).
\end{aligned}$$

Thus, after $K_{\bar{\varepsilon}}$ iterations, the policy induced by the return distribution becomes the optimal policy. Beyond this point, the distributional optimality operator transitions to the distributional Bellman operator for the optimal policy, which is a known $\gamma$-contraction with respect to $\bar{w}_p$. Using this result, we conclude that the combined operator $\Pi_Q \mathcal{T}^{\mathcal{G}_l}$ is a contraction with respect to $\bar{w}_\infty$, as established in Dabney et al. (2018b, Proposition 2).

## F. Additional discussion on Time and Dynamic Consistency

In this section, we need to introduce new notations to discuss the flow of information. Suppose we have a sequence of real-valued random variable spaces denoted as $\mathcal{Z}_0 \subset \cdots \subset \mathcal{Z}_T, \mathcal{Z}_t := L_p(\Omega, \mathcal{F}_t, \mathbb{P})$. Here, $Z_t : \Omega \to \mathbb{R}$ represents an element of the space $\mathcal{Z}_t$.

Moreover, let us define the preference system $\{\rho_{t,T}\}_{t=0}^{T-1}$ as the family of preference mappings $\rho_{t,T} : \mathcal{Z}_T \to \mathcal{Z}_t, t = 0, \ldots, T-1$. The conditional expectation denoted by $\mathbb{E}[\cdot \mid \mathcal{F}_t]$ is an example of such mappings. With these notations, our optimization problem defined in Equation 1 can be written with $\rho = \rho_{0,T}$ and $Z^\pi = Z_{0,T}^\pi$, where $Z_{t,T}^\pi \in \mathcal{Z}_T$ denotes the cumulative reward starting from time $t$. The following definitions help us discuss the connection between the risk measure $\rho$ and the policy $\pi$:

**Definition F.1 (Time-consistency).** An optimal policy $\pi^* = (a_0^*, \ldots, a_T^*)$ is time-consistent if for any $t = 1, \ldots, T$, the shifted policy $\overrightarrow{\pi}^* = (a_t^*, \ldots, a_T^*)$ is optimal for

$$\max_{\pi \in \boldsymbol{\pi}} \rho_{t,T}\left(Z_{t,T}^\pi\right). \tag{35}$$

**Definition F.2 (Dynamic-consistency).** The preference system $\{\rho_{t,T}\}_{t=0}^{T-1}$ is said to exhibit dynamic consistency if the following implication holds for all $0 \leq t_1 < t_2 \leq T-1$:

$$\rho_{t_2,T}(Z) \succeq \rho_{t_2,T}(Z') \implies \rho_{t_1,T}(Z) \succeq \rho_{t_1,T}(Z') \quad Z, Z' \in \mathcal{Z}_T, 0 \leq t_1 < t_2 \leq T-1. \tag{36}$$

Additionally, this preference system is said to exhibit strict dynamic consistency if the following implication holds:

$$\rho_{t_2,T}(Z) \succ \rho_{t_2,T}(Z') \implies \rho_{t_1,T}(Z) \succ \rho_{t_1,T}(Z') \quad Z, Z' \in \mathcal{Z}_T, 0 \leq t_1 < t_2 \leq T-1. \tag{37}$$

Note that dynamic-consistency is a property of a preference system and time-consistency is a property of a policy w.r.t a preference system. Although some authors (e.g. Ruszczyński (2010)) have used the term "time-consistency" for preference

---

[5]For cases with multiple optimal policies in the risk-neutral setting, refer to Bellemare et al. (2023, Section 7.5). Extending this result to the risk-sensitive case is straightforward.

systems, in this context, we maintain the distinction between these two terms. The primary rationale behind this distinction is that using a dynamically consistent preference system implies a time-consistent policy only when the optimal policy is unique. In scenarios with multiple optimal policies, additional conditions must be satisfied (Shapiro et al., 2014). Nonetheless, employing these preference systems has been a widely adopted approach in the RSRL literature to ensure the time-consistency of the optimal policy[6]. To understand the necessary properties of a dynamically consistent preference system, we require additional definitions:

**Definition F.3** (**Recursivity**). The preference system $\{\rho_{t,T}\}_{t=0}^{T-1}$ is said to be recursive if

$$\rho_{t_1,T}\left(\rho_{t_2,T}(Z)\right) = \rho_{t_1,T}(Z) \quad Z \in \mathcal{Z}_T, 0 \le t_1 < t_2 \le T-1. \tag{38}$$

For instance, Kupper & Schachermayer (2009) show that the only law invariant convex risk measure that has the recursion property $\rho\left(\rho(\cdot \mid \mathcal{G})\right) = \rho(\cdot)$ for $\mathcal{G} \subset \mathcal{F}, \mathcal{G} \ne \mathcal{F}$ is the Entropic risk measure:

$$\rho(Z) = \frac{1}{\gamma} \log \mathbb{E}[\exp(\gamma Z)], \gamma \in [0, \infty]. \tag{39}$$

Therefore, using the above-mentioned $\rho$ yields a recursive preference system. The Entropic risk measure is monotone, translation-invariant, and convex. However, it does not have the positive homogeneity property, so it is not suitable for applications in which this property is essential. Nevertheless, the risk measures $\rho(\cdot) = \mathbb{E}(\cdot)$ and $\rho(\cdot) = \operatorname{ess\,sup}(\cdot)$, which are the boundary cases of the Entropic risk measure with $\gamma = 0$ and $\gamma = \infty$, have the positive homogeneity property and therefore are coherent risk measures.

**Definition F.4** (**Decomposability**). The preference mappings $\rho_{t,T}$ are considered to be decomposable via a family of one-step mappings $\rho_t : \mathcal{Z}_{t+1} \to \mathcal{Z}_t$ if they can be expressed as compositions

$$\rho_{t,T}(Z) = \rho_t\left(\rho_{t+1}\left(\cdots \rho_{T-1}(Z)\right)\right), Z \in \mathcal{Z}_T. \tag{40}$$

It is easily seen that the preference mappings of a recursive preference system, such as the one with the Entropic risk measure, are also decomposable. The inverse, however, is not always true and a set of decomposable preference mappings constitute a recursive preference system only if their corresponding one-step mappings are translation-invariant and $\rho_t(0) = 0$ for $t = 0, \ldots, T$. For instance, a convex conditional risk measure such as $\rho_t = \operatorname{CVaR}_\alpha(\cdot \mid \mathcal{F}_t)$ can be used as the one-step mapping and establish a decomposable and recursive preference system[7]. At last, in both of these cases, whether the preference mappings of a recursive preference system or the one-step mappings of a decomposable preference mapping are monotone, the preference system is dynamically consistent.

As mentioned before, the dynamic consistency of the preference system only implies the time-consistency of a unique optimal policy. To guarantee the time-consistency of all optimal policies, Shapiro & Ugurlu (2016) shows that the preference system has to be strictly dynamically consistent. This requires the preference mappings of a recursive preference system or the one-step mappings of a decomposable preference mapping to be strictly monotone, i.e, the following implication must hold:

$$Z \succ Z' \implies \rho_{t,T}(Z) \succ \rho_{t,T}(Z'), Z, Z' \in \mathcal{Z}_T.$$

The Spectral risk measure is an example of strictly monotone preference mappings only if the risk spectrum $\phi(u)$ is positive on the interval $(0, 1)$. Consequently, $\operatorname{CVaR}_\alpha$ is not strictly monotone for $\alpha \in (0, 1)$ and we cannot deduce that, for the preference system characterized by nested Conditional Value-at-Risk, every optimal solution of the corresponding reference problem is time-consistent. An easy way to ensure that $\phi(u) = \int_u^1 \mu(\mathrm{d}\alpha) > 0$ for $u \in (0, 1)$ is to check whether $\mu(\mathrm{d}\alpha)|_{\alpha=1}$ is non-zero or not. In other words, if the risk measure assigns a non-zero weight on the expectation ($\operatorname{CVaR}_1$), the resulting SRM is strictly monotone.

The decomposition theorem shows that the preference mappings $\rho_{t,T}$ can also be provided for SRM. It also shows that these preference mappings are strictly monotone if the initial risk measure is a strictly monotone SRM. This property is evident since for $\alpha = 1$, $\xi^\alpha$ and consequently $\alpha \xi_t^\alpha$ is also 1. Therefore, the weight of $\operatorname{CVaR}_1$ in the preference mappings would

---

[6]In these works, a set of dynamic programming equations are defined and the optimal policies serve as a solution to these equations, which ensure the time-consistency. Additional details on this topic can be explored in the work of Shapiro & Pichler (2016).

[7]These risk measures are also called Nested Risk Measures in the literature

also be non-zero. Intuitively, if the risk measure takes into account the entire distribution to calculate the risk-adjusted value, i.e. has a non-zero weight for the expected value, the resulting preference mappings also have this property.

Additionally, the goal of analyzing the evolution of risk preferences over time can be achieved with the preference mappings, without the need for deriving the one-step mappings. The intermediate random variable $\xi_{t,t-1}$ in the one-step mappings shows how the risk preference at time $t$ changes compared to the previous risk preference at time $t-1$, however, $\xi_t^\alpha$ in the decomposition theorem shows how the risk preference at time $t$ changes compared to the initial risk preference. For example, the CVaR risk level at time $t$, $\alpha_t$, can be written as both $\alpha_{t-1}\xi_{t,t-1}$ or $\alpha\xi_t$. Similarly, in the CVaR case, the risk parameter $B_{t+1}$ can be written as both $(B_t - R_t)/\gamma$ or $(B_0 - S_t)/C_t$.

## G. Proof of Theorem 5.1

For general distributions, we have $\xi^\alpha(z) = 1/\alpha$ for $z < \lambda_\alpha$ and $\xi^\alpha(z) = 0$ for $z > \lambda_\alpha$. To discuss the value of $\xi^\alpha(z)$ when $z = \lambda_\alpha$, let us consider two cases based on the continuity of $F_G(z)$ at $z = \lambda_\alpha$, i.e, whether $p_G(\lambda_\alpha) = 0$ or $p_G(\lambda_\alpha) > 0$. For the first case, we simply have

$$\xi^\alpha(z) = \begin{cases} 1/\alpha & \text{if } z \leq \lambda_\alpha \\ 0 & \text{if } z > \lambda_\alpha \end{cases} \tag{41}$$

Since $G$ is a convex combination of $s_t + c_t G_t$, we know that $p_G(\lambda_\alpha) = 0$ implies $p_{G_t}((\lambda_\alpha - s_t)/c_t) = 0$. Therefore, we have:

$$\begin{aligned}
\xi_t^\alpha &= \mathbb{E}\left[\xi^\alpha \mid \mathcal{F}_t\right] \\
&= \frac{1}{\alpha}\mathbb{E}\left[\mathbb{1}_{\{s_t + c_t G_t \leq \lambda_\alpha\}}\right] \\
&= \frac{1}{\alpha}F_{G_t}(\frac{\lambda_\alpha - s_t}{c_t}) \\
\implies \alpha\xi_t^\alpha &= F_{G_t}(\frac{\lambda_\alpha - s_t}{c_t})
\end{aligned} \tag{42}$$

For the second case where $p_G(\lambda_\alpha) > 0$, we use the fact that $\mathbb{E}[\xi^\alpha] = 1$ to write $\xi^\alpha(\lambda_\alpha)$ as a function of $F_G(\lambda_\alpha)$ and $p_G(\lambda_\alpha)$:

$$\xi^\alpha(z) = \begin{cases} 1/\alpha & \text{if } z < \lambda_\alpha \\ (1 - \frac{1}{\alpha}(F_G(\lambda_\alpha) - p_G(\lambda_\alpha)))/p_G(\lambda_\alpha) & \text{if } z = \lambda_\alpha \\ 0 & \text{if } z > \lambda_\alpha \end{cases} \tag{43}$$

Note that the set $\{z < \lambda_\alpha\}$ can be empty, especially for small $\alpha$. In this case, $\xi^\alpha(\lambda_\alpha) = 1/p_G(\lambda_\alpha)$ would be the only non-zero $\xi^\alpha(z)$. Using this information, we can calculate $\xi_t^\alpha$:

$$\begin{aligned}
\xi_t^\alpha &= \mathbb{E}\left[\xi^\alpha \mid \mathcal{F}_t\right] \\
&= \frac{1}{\alpha}\cdot\mathbb{E}\left[\mathbb{1}_{\{s_t + c_t G_t < \lambda_\alpha\}}\right] + \frac{1 - \frac{1}{\alpha}(F_G(\lambda_\alpha) - p_G(\lambda_\alpha))}{p_G(\lambda_\alpha)}\cdot\mathbb{E}\left[\mathbb{1}_{\{s_t + c_t G_t = \lambda_\alpha\}}\right] \\
&= \frac{1}{\alpha}(F_{G_t}(\frac{\lambda_\alpha - s_t}{c_t}) - p_{G_t}(\frac{\lambda_\alpha - s_t}{c_t})) + \frac{1 - \frac{1}{\alpha}(F_G(\lambda_\alpha) - p_G(\lambda_\alpha))}{p_G(\lambda_\alpha)}\cdot p_{G_t}(\frac{\lambda_\alpha - s_t}{c_t}) \\
&= \frac{1}{\alpha}F_{G_t}(\frac{\lambda_\alpha - s_t}{c_t}) - \frac{1}{\alpha}p_{G_t}(\frac{\lambda_\alpha - s_t}{c_t})\cdot\left(1 - \frac{\alpha - (F_G(\lambda_\alpha) - p_G(\lambda_\alpha))}{p_G(\lambda_\alpha)}\right) \\
&= \frac{1}{\alpha}F_{G_t}(\frac{\lambda_\alpha - s_t}{c_t}) - \frac{1}{\alpha}p_{G_t}(\frac{\lambda_\alpha - s_t}{c_t})\cdot\frac{F_G(\lambda_\alpha) - \alpha}{p_G(\lambda_\alpha)} \\
\implies \alpha\xi_t^\alpha &= F_{G_t}(\frac{\lambda_\alpha - s_t}{c_t}) - p_{G_t}(\frac{\lambda_\alpha - s_t}{c_t})\cdot\frac{F_G(\lambda_\alpha) - \alpha}{p_G(\lambda_\alpha)}
\end{aligned} \tag{44}$$

Notice that this can also be simplified to

$$\alpha\xi_t^\alpha = F_{G_t}(\frac{\lambda_\alpha - s_t}{c_t}) \tag{45}$$

if $F_{G_t}(z)$ does not have a discontinuity at $z = \frac{\lambda_\alpha - s_t}{c_t}$.

In the DRL framework, only estimates of the return-distributions are available. When estimating the distribution with the Quantile representation, it's easy to see that $\lambda_\alpha = \theta_i$ for $\tau_{i-1} \leq \alpha < \tau_i$, so $p_G(\lambda_\alpha) = 1/N$. If $(\lambda_\alpha - s_t)/c_t$ is equal to any of the estimated $\theta_{t,i}$, we have $p_{G_t}((\lambda_\alpha - s_t)/c_t) = 1/N$ and $\alpha\xi^\alpha$ can be estimated with

$$\alpha\xi_t^\alpha = F_{G_t}(\frac{\lambda_\alpha - s_t}{c_t}) - (F_G(\lambda_\alpha) - \alpha).$$

Otherwise, we have $p_{G_t}((\lambda_\alpha - s_t)/c_t) = 0$ and $\alpha\xi^\alpha$ can be estimated with Equation 45.

## H. Examples for Calculating the Intermediate Risk Preferences

**Example 2.** Suppose that the quantiles of $G$ and $G_t$, denoted by $\theta$ and $\theta_t$, are given as in the table below. Also suppose that $s_t = 5$, $c_t = 0.8$, and we are interested in the following risk measure at the initial state:

$$\rho(G) = 0.6 \cdot \text{CVaR}_{0.25}(G) + 0.4 \cdot \text{CVaR}_{0.8}(G).$$

Table 4: The quantiles of $G$ and $G_t$, and the dual variables $\xi^{0.25}$ and $\xi^{0.8}$ to calculate $\rho(G)$

| $\hat{\tau}$ | $\theta$ | $\xi^{0.25}$ | $\xi^{0.8}$ | $\theta_t$ |
|---|---|---|---|---|
| 5% | 7 | 4 | 1.25 | 5 |
| 15% | 9 | 4 | 1.25 | 6 |
| 25% | 12 | 2 | 1.25 | 8 |
| 35% | 20 | 0 | 1.25 | 14 |
| 45% | 21 | 0 | 1.25 | 15 |
| 55% | 27 | 0 | 1.25 | 17 |
| 65% | 30 | 0 | 1.25 | 21 |
| 75% | 32 | 0 | 1.25 | 25 |
| 85% | 39 | 0 | 0 | 28 |
| 95% | 46 | 0 | 0 | 35 |

For $\alpha = 0.25$, $\text{CVaR}_\alpha$ and $\lambda_\alpha$ are 8.8 and 12. For $\alpha = 0.8$, these values are 19.75 and 39. Now we can use Equation 45 to calculate the $\alpha\xi_t^\alpha$ values.

$$0.25\xi_t^{0.25} = F_{G_t}(\frac{12 - 5}{0.8}) = F_{G_t}(8.75) = 0.3,$$

$$0.8\xi_t^{0.8} = F_{G_t}(\frac{39 - 5}{0.8}) = F_{G_t}(42.5) = 1.0.$$

With $\xi_t^{0.25} = 0.3/0.25 = 1.2$ and $\xi_t^{0.8} = 1.0/0.8 = 1.25$, we can see that at time $t$, the risk measure changes to

$$\rho_\xi(G_t) = 0.59 \cdot \text{CVaR}_{0.3}(G_t) + 0.41 \cdot \text{CVaR}_{1.0}(G_t).$$

where $\xi = 0.6 \cdot 1.2 + 0.4 \cdot 1.25 = 1.22$.

**Example 3.** In this example, we provide an intuition behind the computation of intermediate risk preferences in the Mean-reversion Trading environment. To start, let's examine a sample trajectory within our environment, where we employ our model optimized for $\text{CVaR}_{0.5}$. The details of this trajectory can be found in Table 5. In Figure 6(a), we also present the return distribution $G(x_t, s_t, c_t, a_t)$ at each time step $t$. These distributions characterize the agent's future reward when it begins in the state-action pair $(x_t, s_t, c_t, a_t)$ at time $t$ and follows the optimal policy.

In our model, the risk preference of the agent chosen at time 0, which is associated with function $h$, remains constant throughout the trajectory. For instance, in our specific scenario with $\lambda_{0.5} = 0.874$, the agent's action selection is based on the average return below this value, corresponding to the 0.5-quantile of the return distribution at the initial state. In order to apply this risk preference in all subsequent states, we need to align the return distribution in those states with the agent's

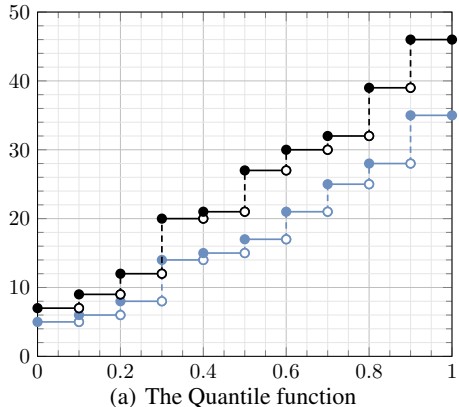
(a) The Quantile function

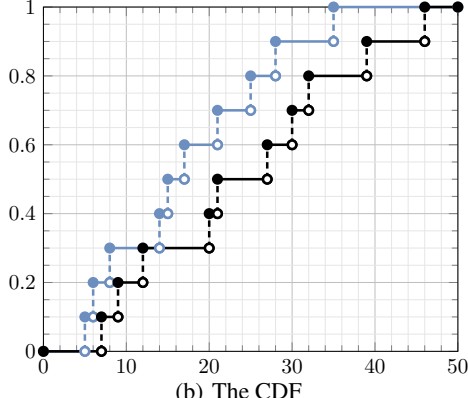
(b) The CDF

Figure 5: The Quantile function and the CDF of $G$ (black) and $G_t$ (blue) in Example 2

Table 5: The states and actions of a single trajectory in our algorithmic trading environment

| $t$ | 0 | 1 | 2 | 3 | 4 | 5 | 6 | 7 | 8 | 9 | 10 |
|---|---|---|---|---|---|---|---|---|---|---|---|
| $P_t$ | 1.000 | 0.606 | 0.768 | 1.053 | 0.796 | 0.934 | 0.569 | 0.636 | 0.238 | 0.698 | 0.870 |
| $q_t$ | 0.000 | -0.400 | 1.600 | 1.800 | -0.200 | 0.200 | 0.600 | 1.000 | 0.400 | 0.800 | 0.200 |
| $s_t$ | 0.000 | 0.399 | -0.820 | -0.971 | 1.053 | 0.746 | 0.390 | 0.175 | 0.529 | 0.440 | 0.962 |
| $c_t$ | 1.000 | 0.990 | 0.980 | 0.970 | 0.961 | 0.951 | 0.941 | 0.932 | 0.923 | 0.914 | 0.904 |
| $a_t$ | -0.400 | 2.000 | 0.200 | -2.000 | 0.400 | 0.400 | 0.400 | -0.600 | 0.400 | -0.600 | 0.000 |
| $r_t$ | 0.399 | -1.232 | -0.154 | 2.086 | -0.319 | -0.374 | -0.228 | 0.380 | -0.096 | 0.571 | 0.000 |
| $\alpha_t$ | 0.500 | 0.168 | 0.355 | 0.083 | 0.089 | 0.147 | 0.544 | 0.702 | 0.819 | 0.084 | 0.000 |

perspective at the initial time. This alignment is achieved by scaling the return distribution by $c_t$ and adding $s_t$, as illustrated in Figure 6(b).

Now we can see that the value $\lambda_{0.5} = 0.874$ corresponds to a different quantile of the return distribution in subsequent states. For instance, action selection w.r.t the $0.5$-quantile of the return distribution at time $0$ shifts to $0.168$-quantile of the return distribution at time $1$. This mechanism enables us to observe how the agent's risk preference evolves over time. Here, we demonstrated the process for a single $\alpha$, but more complicated SRMs follow similar steps. The only additional step would be the calculation of the weight of each component of the risk measure, similar to the example in Appendix 2.

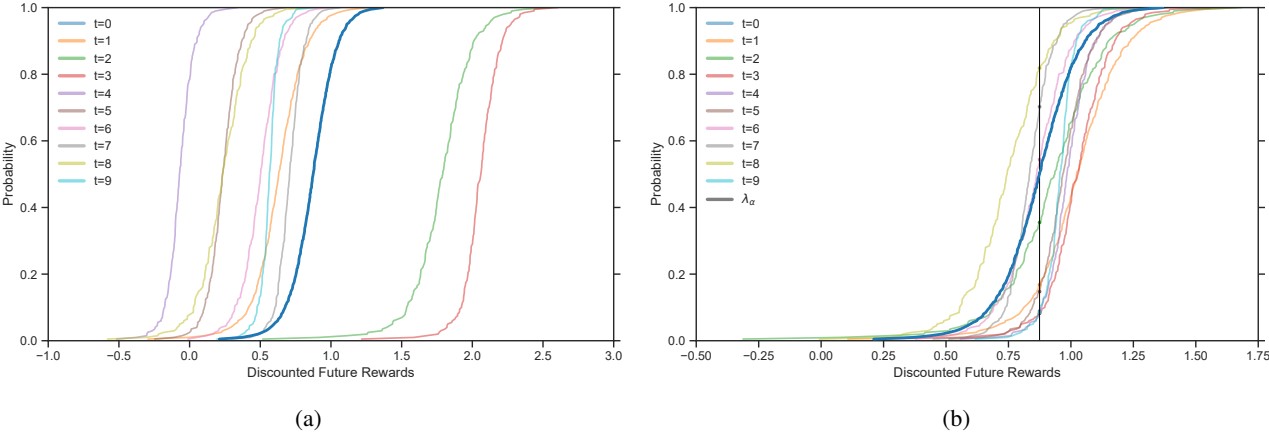

(a)                                                      (b)

Figure 6: Figure 6(a) and 6(b) illustrate the CDF of $G(x_t, s_t, c_t, a_t)$ and $s_t + c_t G(x_t, s_t, c_t, a_t)$ for each state-action pair in a trajectory.

# I. Experiments with number of Quantiles

Due to the approximation of the probability measure $\mu$, an important question arises: Can our model find a policy that maximizes the expected return, the primary objective of the risk-neutral QR-DQN algorithm? To address this question, we conducted a comparison of the expected return produced by our model under varying quantile numbers ($N$), with $\tilde{\mu}_N$ set to 1, in the mean-reversion trading example. The results of this experiment, presented in Figure 7(a), demonstrate that as the number of quantiles increases, our model not only matches the performance of the risk-neutral algorithm but surpasses it, yielding superior expected returns. Furthermore, in Figure 7(b), we observe that the improvement extends beyond expected returns. The policy derived from our algorithm consistently attains higher $\mathrm{CVaR}_\alpha$ values for all $\alpha \in (0, 1]$.

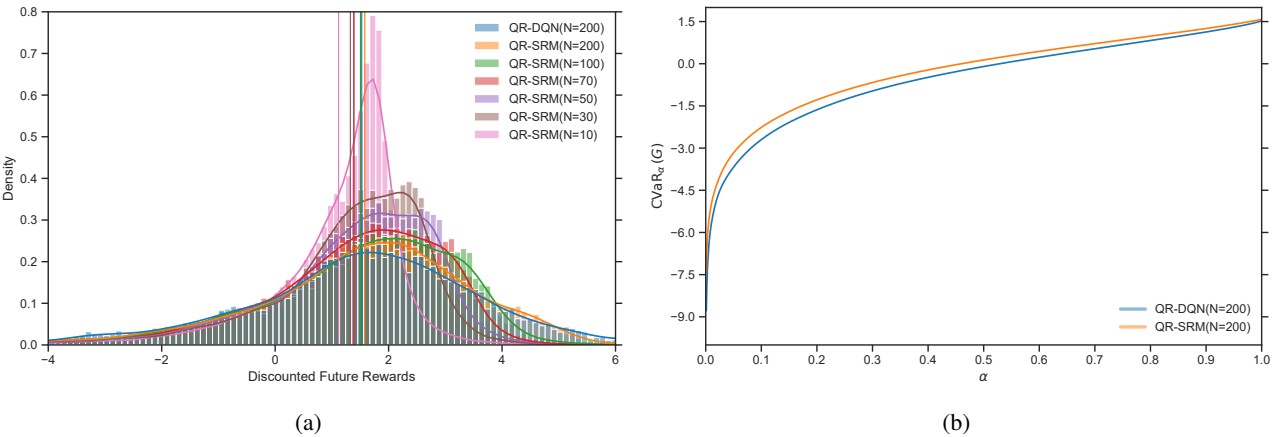

(a)            (b)

Figure 7: Figure 7(a) displays the distribution of Cumulative Discounted Rewards for policies with different number of quantiles. The solid lines in this figure represent $\mathbb{E}[G]$. Figure 7(b) compares the performance of QR-SRM($\alpha$=1) against QR-DQN, both with 200 quantiles, w.r.t to $\mathrm{CVaR}_\alpha$ for all $\alpha \in (0, 1]$.

However, this enhanced performance comes at a cost. When normalizing all models' scores with the QR-DQN($N$=50) score, as depicted in Figure 8(a), all models reach an equivalent performance level within the same number of steps[8]. Yet, this figure can be misleading since the time of action selection at each step increases quadratically with the number of quantiles. Specifically, for an action space size denoted as $A$, the QR-DQN model's action selection requires $\mathcal{O}(AN)$ operations, in contrast to our model, which requires $\mathcal{O}(AN^2)$ operations. Figure 8(b) presents the score plotted against the training time normalized to the training time of the QR-DQN($N$=50), revealing that transitioning from 50 quantiles to 200 has a less pronounced impact on the QR-DQN model compared to our model.

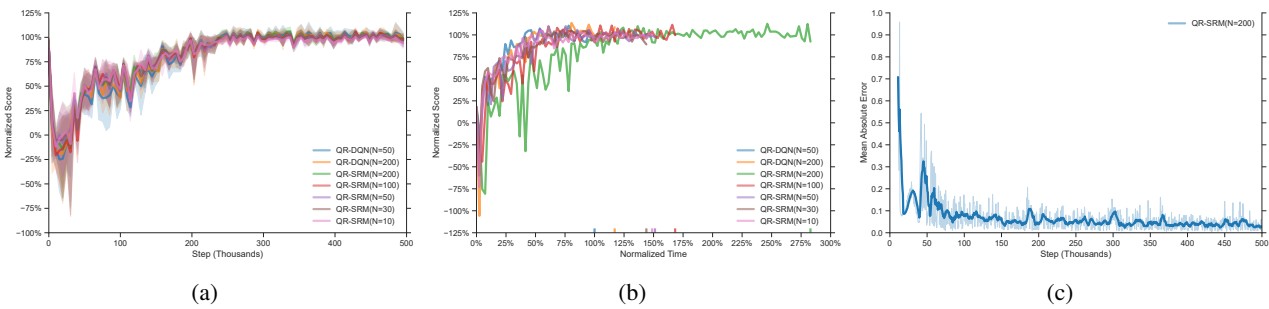

(a)            (b)            (c)

Figure 8: Figure 8(a) and 8(b) displays the moving expected reward of our model and the QR-DQN model with different numbers of quantiles, plotted against the number of steps and time. Figure 8(c) shows the mean absolute error between consecutive estimations of the return distribution $G(x_0, 0, 1, a_0^*)$.

In our algorithm, the estimation of function $h$ is updated periodically. This estimation is directly linked to the estimation of

---

[8]Each step corresponds to a single interaction of the agent with the environment.

$G(x_0, 0, 1, a_0^*)$, making the convergence of this return distribution a useful indicator for the convergence of the function $h$. Figure 8(c) visualizes this convergence, presenting the mean absolute error between consecutive estimations of the return distribution.

## J. Details of the Environments

### J.1. American Put Option Trading

In this environment, we assume that the price of the underlying asset follows a Geometric Brownian Motion, characterized by the differential equation $\mathrm{d}P_t = \zeta P_t \mathrm{d}t + \sigma P_t \mathrm{d}W_t$, where $\zeta = -0.3$ is the drift, $\sigma = 0.3$ is the volatility, the initial price is $P_0 = 1$, and $W_t$ is a standard Brownian motion. The strike price of the put option is assumed to be $K = 1$. At each time step, the agent can either exercise the option and receive $r_t = \max\{0, K - P_t\}$ or hold the option to receive a reward at future steps. At maturity, if the option hasn't been exercised yet, the agent automatically receives $r_T = \max\{0, K - P_T\}$.

### J.2. Mean-reversion Trading Strategy

In this algorithmic trading framework, the asset price follows an Ornstein-Uhlenbeck process, characterized by the differential equation $\mathrm{d}P_t = \kappa(\zeta - P_t)\mathrm{d}t + \sigma \mathrm{d}W_t$, where $\zeta = 1$ is the long-term mean level, $\kappa = 2$ determines the speed of reversion to mean. At each time step $t = 0, \cdots, T - 1$, the agent takes an action $a_t \in (-a_{\max}, a_{\max})$, corresponding to trading quantities of the asset and changes its inventory $q_t \in (-q_{\max}, q_{\max})$. The reward is defined as $r_t = -a_t P_t - \varphi(a_t)^2$ for $0 \leq t \leq T - 2$ and $r_{T-1} = -a_{T-1}P_{T-1} - \varphi(a_{T-1})^2 + q_T P_T - \psi q_T^2$ for the final time step. Here, $\varphi = 0.005$ represents the transaction cost and $\psi = 0.5$ signifies the terminal penalty. In our setup, the agent faces penalties for holding any assets at the final time step $T$. Consequently, the reward at time step $T - 1$ has an additional term for the agent's inventory at time step $T$. In our example, we consider $T = 10$, $q_{\max} = 5$, $a_{\max} = 2$, $\gamma = 0.99$, and discretize the action space into 21 actions.

### J.3. Windy Lunar Lander

The Lunar Lander environment is a classic rocket trajectory optimization problem, involving an 8-dimensional state space and four actions: firing the left or right orientation engines, firing the main engine, or doing nothing. To introduce stochasticity, we enable the wind option. The objective is to guide the lander from the top of the screen to the landing pad. Successful landings yield around 100–140 points. If the lander moves away from the pad, it loses points, while a crash results in an additional penalty of -100 points. Landing safely adds a bonus of +100 points, and each leg that makes contact with the ground earns +10 points. Firing the main engine incurs a penalty of -0.3 points per frame, while firing the side engines costs -0.03 points per frame.

## K. Implementation details

This section provides an overview of the implementation details of our model. We adopt the single-file implementation of RL algorithms from CleanRL (Huang et al., 2022) for clarity. In this approach, the model and its training are encapsulated within a single file. The code for the project is available at `https://github.com/MehrdadMoghimi/QRSRM`.

Table 6: Default hyperparameters in different models

| Hyperparameter | Value |
|---|---|
| Learning Rate | 2.5e-4 |
| Discount Factor ($\gamma$) | 0.99 |
| Batch Size | 256 |
| Number of Quantiles | 50 |

The repository contains four Python files for each algorithm discussed in section 6 and Appendix I. The `qrsrm.py` file defines the state-action value function with a feed-forward network that takes $(X, S, C)$ as input and outputs a $N \times A$ dimensional vector representing the quantile function of all actions. This neural network comprises three hidden layers, each with 128 neurons. The value function is similar across other files, with the only difference being their input value, which can be $(X)$ in `qrdqn.py` and `qricvar.py` or $(X, B)$ in `qrcvar.py`.

In `qrcvar.py` and `qricvar.py`, the variable `alpha` determines the risk preference of the agent. In `qrsrm.py`, the user can choose between different risk measures with the `risk-measure` variable. The value of `CVaR`, `WSCVaR`, `Dual`, and `Exp` for this variable is associated with the CVaR, Weighted sum of CVaRs, Dual Power and Exponential risk measures.

The number of timesteps to train each algorithm is determined by `total-timesteps` variable. A fraction of these timesteps is allocated to $\epsilon$-greedy exploration and the rest is allocated to learning the value function accurately. Also, in `qrsrm.py` and `qrcvar.py`, the estimation of function $h$ and target value $b$ needs frequent updating. The value of variables `h_frequency` and `b_frequency` in these files determine the update frequency for these estimations. Lastly, techniques such as Replay Buffers and Target Networks are employed to stabilize the training process for all of the algorithms.

The custom environments used in our experiments are available in the `custom_envs.py` file. We implemented the American Option Trading and Mean-reversion Trading environments using the Gymnasium (formerly OpenAI Gym) package (Towers et al., 2023). This package allows for the definition of the state space, action space, and environment dynamics with simple functions. The primary function is the `step` function, which takes an action as input and outputs the reward and the next state based on the current state.

The state space augmentation for QR-SRM and QR-CVaR models is also defined using two environment wrappers. These wrappers automatically store the target value $B$ or the accumulated discounted reward $S$ and the discount factor $C$ for a trajectory. The key advantage of these wrappers is their compatibility with any environment available in the Gymnasium package.

