# OpenReview forum: "Beyond CVaR: Leveraging Static Spectral Risk Measures for Enhanced Decision-Making in Distributional Reinforcement Learning"
_ICML.cc/2025/Conference — ICML 2025 poster_

### Official Review · Reviewer_eG5t · 2025-02-18

**Overall Recommendation:** 2

**Summary:**

This paper studies how to optimize a single static Spectral Risk Measure (SRM) for an RL agent, leveraging a distributional RL framework. By extending the state to track discounted returns and employing quantile-based updates, the authors propose a two-tier algorithmic approach:

Inner Optimization: Fix an approximate function $h$ related to the SRM, then learn a greedy policy in the extended MDP.
Outer Optimization: Update $h$ using closed form solution to better match the (new) return distribution of the initial state.

The authors highlight that using a single SRM generalizes beyond basic CVaR-based methods, potentially offering more flexibility in risk profiles. Experiments on algorithmic trading tasks and a Windy Lunar Lander environment suggest that their approach can produce policies that better align with static SRMs compared to baselines such as standard CVaR-based methods or purely risk-neutral approaches.

**Claims And Evidence:**

1.
- Claim: The paper argued that it can directly optimize a spectral risk measure (broader than CVaR) through a distributional approach, thereby obtaining a single “best” policy according to that measure.
- Evidence: Empirical demonstrations on trading tasks show that the learned distribution of returns indeed improves under the selected SRM.

2.
- Claim: The method achieves a “monotonic” or “non-decreasing lower bound” style improvement in the outer loop.
- Evidence: The authors prove a type of asymptotic convergence result for the policy improvement step (inner optimization). However, they do not provide finite-step performance bounds or sample-complexity analysis.

**Essential References Not Discussed:**

1. Kim, Dohyeong, et al. "Spectral-Risk Safe Reinforcement Learning with Convergence Guarantees." Advances in neural information processing systems 2024.
- While Kim et al. use SRM in a constraint setting (safe RL) and this paper uses SRM as a single-objective optimization, the overall methods (bilevel structure, state augmentation) are very similar. Yet the paper does not explicitly discuss or compare its approach to Kim et al.
- Add a dedicated subsection contrasting this method with Kim et al.
- Emphasize where the approaches are methodologically parallel (e.g., the idea of rewriting the SRM using supremum representation; the linearization of risk measure in the inner optimization).
- Pinpoint novel aspects in this paper, such as the closed-form solution in Eq. (6) vs. the optimization-based approach for the supremum in Kim et al. (Eq. 5 in this work).

2. Advanced Distributional RL: If your approach relies on or extends any of the standard distributional algorithms (C51, QR-DQN, IQN, etc.), referencing these helps readers see how your design decisions compare or could be integrated with more sophisticated distributional frameworks (e.g., FQF).

2-1. Dabney, Will, et al. "Implicit quantile networks for distributional reinforcement learning." International conference on machine learning. PMLR, 2018.
- Extends quantile ideas via implicit quantile networks (IQN).

2-2. Yang, Derek, et al. "Fully parameterized quantile function for distributional reinforcement learning." Advances in neural information processing systems 32 (2019).
- Improves upon quantile-based DRL by parameterizing quantiles themselves as learnable functions.

**Experimental Designs Or Analyses:**

1. Tasks:
- Two trading-like environments (American option, mean-reversion) demonstrate the algorithm’s ability to favor certain parts of the distribution (e.g., different quantile levels).
- Windy Lunar Lander adds complexity but remains a somewhat “toy” environment compared to many large-scale or safety-critical domains.

2. Comparisons:
- Against risk-neutral QR-DQN, a fixed-CVaR version, and “iCVaR” from prior work. Results show each approach yields distinct return distributions, with the SRM-based approach often outdoing purely CVaR or risk-neutral baselines in the chosen metric.

3. Comments:
- The tasks could be considered simple or proof-of-concept. For a deep distributional RL method, it would be insightful to test more safety-critical or complex environments (e.g., from Safety Gymnasium: https://safety-gymnasium.readthedocs.io/en/latest/index.html).
- In the trading tasks, standard finance metrics (mean-variance, Sharpe ratio, maximum drawdown) are absent, making it less clear how these policies compare to typical trading strategy benchmarks.

**Methods And Evaluation Criteria:**

Methods:
- The extended MDP approach (tracking discounted returns in the state) is standard for risk-based RL.
- The inner optimization uses quantile RL updates (in the style of QR-DQN).
- The outer optimization updates the SRM parameter $h$ in closed form (from the Kusuoka representation).

Evaluation:
Experiments focus on the resulting return distributions of policy, both visually and via risk metrics (CVaR, exponential risk, etc.). The main shortcoming is that tasks are relatively toy-scale or moderately sized, and the baselines are fairly limited (primarily QR-DQN or variations of CVaR). For a final measure of practical performance, more rigorous tasks and additional metrics (especially in trading, e.g., Sharpe ratio, drawdown) would strengthen credibility.

**Other Comments Or Suggestions:**

- Multi-risk or multi-constraint extension: Address the possibility of combining multiple SRMs or constraints to handle different aspects of risk.
- Deeper experimental demonstration (including standard trading metrics, more complicated tasks) would enhance confidence in the method’s real-world applicability.

**Other Strengths And Weaknesses:**

1. Strengths:
- Demonstrates clear expansions beyond pure CVaR to a flexible set of SRMs.
- Good interpretability argument for static risk measures, using a decomposition approach.
- Shows how distributional RL can incorporate these measures with (arguably) minimal overhead.

2. Weaknesses:
- Limited comparison to Kim et al. despite using very similar core techniques.
- Convergence analysis does not address finite-sample complexities.
- Single-objective viewpoint: The final converged measure is still a single SRM, which might not serve multiple stakeholder or multi-risk contexts.
- Experiments remain modest and do not thoroughly test the approach in safety-critical or large-scale RL scenarios.

**Questions For Authors:**

1. Could you explicitly discuss how your approach (bilevel with a closed-form outer step) differs in practice from Kim et al.’s primal-dual approach, which also uses state augmentation and spectral risk?

[Kim et al.] Kim, Dohyeong, et al. "Spectral-Risk Safe Reinforcement Learning with Convergence Guarantees." Advances in neural information processing systems 2024.

2. Is there a straightforward way to bound approximation errors or obtain a sample-complexity result for the inner–outer loops?

3. You mention that the risk measure $h$ can “change” mid-training (mentioned in Introduction), but eventually converges to a single SRM. Could you clarify whether, at convergence, the agent is strictly tied to one final risk measure? In other words, do you expect any meaningful “variety” in the agent’s risk preference by the end of training, or is it effectively just a single risk measure?

3-1. If a policy is optimized for one particular SRM, it may perform poorly under different risk criteria (e.g., a strong CVaR policy but weak expected return). Have you considered how users might adapt your method if they wish to account for multiple forms of risk or multiple stakeholder objectives simultaneously?

4.  How does this policy compare when measured by classical trading benchmarks such as mean–variance (Markowitz), the Sharpe ratio, or maximum drawdown? Could you include these metrics to show practical relevance?

**Relation To Broader Scientific Literature:**

- Spectral Risk Measures: The approach is consistent with the Kusuoka or supremum representation of coherent risk measures.
- Distributional RL: The method leverages standard quantile approximation (QR-DQN) and is part of a growing body of risk-oriented distributional methods.

**Theoretical Claims:**

1. Asymptotic Convergence of Inner Loop: The paper shows that for the inner loop (policy learning under a fixed $h$), the linearization of the sub-risk measure allows for a standard policy-improvement argument in distributional RL. This is reminiscent of known results (e.g., Kim et al. 2023, which obtains a similar linearization in a constrained RL setting). Furthermore, the paper only proves a monotonic lower bound in the tabular or exact distribution scenario. Under function approximation, or in finite-sample contexts, rigorous finite-time or non-asymptotic analyses are not provided.

2. Monotonic Outer Updates: By iteratively re-estimating $h$ from the newly observed initial-state return distribution, the algorithm is said to climb toward an optimal measure. To the best of my knowledge, this part is novel in risk-aware RL with SRM.

---

> ### Author Rebuttal · Authors · 2025-03-31
>
> Thank you for recognizing the strengths of our work. We address your questions below.
>
> **Q1:**
>
> We thank the reviewer for this insightful comment. We agree that an explicit comparison with [1] would improve the clarity of our contribution. While both works share methodological components, such as the use of SRMs, a bilevel structure, and state augmentation, they address fundamentally different problems and employ distinct technical approaches.
>
> [1] focuses on risk-constrained RL, where SRMs are used to define constraints. Their method uses a primal-dual optimization approach, requiring gradient-based updates for both the policy and the dual variables associated with the risk constraint. In contrast, our paper addresses risk-sensitive RL, where the SRM is used as the objective function. Our method leverages the SRM’s supremum representation to derive a closed-form outer update (Eq. 6), avoiding a computationally expensive optimization step.
>
> Practically, this leads to three key distinctions:
> - **Computational Efficiency:** Our method eliminates costly outer-loop optimization over dual variables. Instead, the outer update step uses the learned return distribution in the closed-form solution.
>
> - **Theoretical Differences:** Unlike the actor-critic framework in [1], our approach is value-based, requiring a different convergence analysis.
>
> - **Interpretability:** We provide insight into the agent’s intermediate risk preferences through SRM decomposition, an aspect not addressed in [1].
>
> We will include a dedicated subsection summarizing these differences and explicitly reference [1] in the revised manuscript.
>
> **Q2:**
>
> We appreciate this thoughtful comment. Our convergence analysis focuses on asymptotic behavior, demonstrating that QR-SRM monotonically improves a lower bound on the SRM objective (Theorem 4.1). While our work emphasizes practicality and interpretability, other studies have explored the theoretical foundations of spectral risk measures. For instance, [2,3] provide regret and sample complexity bounds for SRM in distributional RL. Additionally, the approximation error for quantile TD algorithms, discussed in Proposition 19 of [4], is directly relevant to our inner optimization step, as we employ the same distributional Bellman operator in the augmented MDP.
>
> **Q3:**
>
> Thank you for raising this important point. Our method indeed optimizes the policy with respect to a single, user-specified SRM, and the final policy at convergence is aligned with that specific risk preference. Thus, after convergence, the agent is strictly tied to that SRM. We do not claim that the learned policy generalizes across all risk criteria, and we agree that a policy optimized for one SRM (e.g., CVaR) may underperform under others (e.g., expected value).
>
> However, the mention of changing risk preferences over time (not during training) refers to a key interpretability contribution of our work: through the decomposition of SRM (Section 5), we are able to analyze the agent's intermediate, state-dependent risk preferences as the return distribution evolves. While the objective remains fixed, the conditional risk levels and weights (e.g., combinations of CVaRs) change across states and over time, providing insight into how the policy’s effective behavior adapts over time. This decomposition does not imply that the algorithm optimizes for multiple SRMs, but rather enhances interpretability within a single SRM objective.
>
> To address multiple risk criteria or stakeholder objectives, we agree this is an interesting and practically relevant direction. One possible extension would involve optimizing for a mixture of SRMs, for example, via multi-objective RL. Some relevant works, such as [5], have explored these directions. Another direction could involve conditioning the policy on the risk parameter. While we do not explore these extensions in this work, we believe our framework provides a strong foundation for such adaptations, and we appreciate the reviewer highlighting this point.
>
> **Q4:**
>
> Thank you for the suggestion. Our work focuses on developing a general RL algorithm based on SRMs, rather than a financial trading strategy. While metrics like mean-variance, Sharpe ratio, and maximum drawdown are important in finance, they are not directly applicable to our benchmark environments. Nonetheless, applying our method to financial data and evaluating it under these metrics is a promising direction for future work.
>
> **References:**
>
> [1] Kim et al. Spectral-Risk Safe Reinforcement Learning with Convergence Guarantees
>
> [2] Bastani et al. Regret Bounds for Risk-Sensitive Reinforcement Learning
>
> [3] Chen et al. Provable Risk-Sensitive Distributional Reinforcement Learning  with General Function Approximation
>
> [4] Rowland et al. An Analysis of Quantile Temporal-Difference Learning
>
> [5] Moffaert et al. Risk-sensitivity through multi-objective reinforcement learning

---

### Official Review · Reviewer_pk66 · 2025-02-21

**Overall Recommendation:** 4

**Summary:**

This paper proposes a distributional RL algorithm for optimizing the Spectral Risk Measures (SRM, with CVar as a special case) of return in discounted MDPs.
Specifically, this paper makes use of a variational representation of SRM proposed by Kusuoka, thus transforming the optimization of SRM into a two-layer optimization problem. The inner-layer optimization problem is almost the same as the problem encountered when optimizing CVaR. It is a problem of solving the optimal risk-neutral strategy for an MDP with augmented states using the current cumulative reward (stock) and discount rate (while also requiring the estimation of the return distribution). As for the outer-layer optimization problem (i.e., the variational representation), it has an explicit solution expressed in terms of the quantiles of the return distribution.
The paper proves the theoretical convergence of the algorithm (assuming that each step can be solved precisely) and characterizes the temporal decomposition of SRMs for the interpretation of the learned policy.
This paper verifies through extensive experiments that this algorithm can optimize the SRM and outperform the previous risk-neutral and risk-sensitive algorithms.

**Claims And Evidence:**

yes

**Essential References Not Discussed:**

As far as I know, no key references are missed.

**Experimental Designs Or Analyses:**

yes

**Methods And Evaluation Criteria:**

yes

**Other Comments Or Suggestions:**

Line 110 reward->return (or cumulative reward)
Line 165 rho_{\tilde{\xi}} is used before definition.

The notations are misleading, for example, in Eqn (7), H in pi_H is history; but h\in\mathcal{H} is concave function, and pi_h is also used. It would be better to use different letters. And using \mathcal{S} to denote [G_min, G_max] is also strange in RL literature.

**Other Strengths And Weaknesses:**

Strength:
1. The generalization from CVaR to SRM is meaningful, and using distributional RL to obtain the explicit solution of the variational representation of SRM is a nice and natural idea.
2. The paper is well-written and the technical parts are not hard to follow.
3. The paper also has extensive experiments to verify the effectiveness of the proposed algorithm.

Weakness:
1. In the right part of line 117-132, the paper introduced non-stationary Markov policy $\Pi_M\subset\Pi_H$ (with stock augmentation), but did not state that $\sup_{\pi\in\Pi_H}J(\pi, h)=\sup_{\pi\in\Pi_M}J(\pi, h)$, and in the following, the paper only focused on $\sup_{\pi\in\Pi_M}J(\pi, h)$. I know that $\sup_{\pi\in\Pi_H}J(\pi, h)=\sup_{\pi\in\Pi_M}J(\pi, h)$ is absolutely correct, but it would be better to clarify it in the paper.
2. Why in Algorithm 1 h_0 is randomly initialized, but in the proof of the convergence of inner problem (Line 649) $h_l$ is $\phi(0)$-Lipschitz, which is not stated in any other places. As far as I know, for such distributional value iteration algorithm (inner optimization problem), the Lipschitz property of the function is necessary to obtain the $\gamma^k$ convergence rate. I think it should be clarified in the paper by e.g. further explaining Eqn.(6).
3. A minor point: This paper pays too much attention to the technical part. I think it would be better to clarify why  generalizing from CVaR to SRM is significant, such as providing and explaining simple and intuitive examples of SRM other than CVaR (line 330-337 present some examples of SRM without enough explanation).

**Questions For Authors:**

I have no question

**Relation To Broader Scientific Literature:**

This paper extends the optimizing the CVaR [1] of return of discounted MDPs with distributional RL to optimizing SRM. (Optimizing SRM in MDP was also considered in [2] without using distributional RL)

[1] Bellemare, M. G., Dabney, W., and Rowland, M. Distributional Reinforcement Learning
[2] Nicole Bauerle and Alexander Glauner. Minimizing spectral risk measures applied to Markov decision processes

**Theoretical Claims:**

yes

---

> ### Author Rebuttal · Authors · 2025-03-31
>
> Thank you for your positive feedback and for recognizing the strengths of our work. We are glad that you found our paper well-written. We address your questions below.
>
> **Q1:**
>
> Thank you for pointing this out. Theorem 2 of [1] and Theorem 3.1 of [2] provide a detailed discussion on the equivalence of these two supremums. We will clarify this in the revised manuscript.
>
> **Q2:**
>
> We appreciate the reviewer’s careful observation. Indeed, the convergence proof of the inner optimization (Line 649) assumes that $h_l$ is $\phi(0)$-Lipschitz. While Algorithm 1 states that $h_0​$ is randomly initialized, we clarify that the required Lipschitz property holds from the beginning.
>
> Specifically, in Algorithm 1, $h_0$ takes the form of Eqn. (6), which utilizes the quantiles of a return distribution and the risk spectrum $\phi$. As detailed in Appendix D, this equation is by construction $\phi(0)$-Lipschitz. Therefore, using the quantiles of a random return distribution ensures that the Lipschitz condition is satisfied for $h_0$.
>
> We acknowledge that this point was not clearly stated in the current version of the paper. In the revision, we will explicitly clarify this in Section 4 and in the algorithm description to avoid confusion.
>
> **Q3:**
>
> As highlighted in our paper’s contributions, SRM provides valuable flexibility for practitioners. For example, Mean-CVaR is widely used in finance, particularly in portfolio management and insurance pricing. While CVaR, as a subclass of SRMs, offers some flexibility through the alpha parameter for adjusting policies, SRM provides even greater flexibility in defining objectives. For instance, WSCVaR allows practitioners to combine multiple CVaR objectives with arbitrary weights.
>
> A key advantage of SRM over CVaR is its adaptability in cases where the ideal risk-sensitive objective is not immediately clear. In certain environments, incorporating risk sensitivity leads to policies that improve both worst-case return and average return, demonstrating that optimizing for risk-sensitive performance does not always come at the cost of lower average returns. By treating the parameters of the risk-sensitive objective as hyperparameters, users can tune them and compare the resulting policies. This flexibility is especially useful in environments where reward models are arbitrarily designed and lack a clear real-world interpretation. For example, while a portfolio manager may have a well-defined objective in a trading environment, such clarity is often absent in environments like Lunar Lander. In these cases, tuning risk objectives as hyperparameters provides a practical solution.
>
> In the experimental section, Table 2 shows that our algorithm with a CVaR objective matches the performance of QR-CVaR from [3], which benefits from stronger convergence guarantees. Furthermore, our algorithm successfully optimizes various risk measures, such as the dual power risk measure and Mean-CVaR, consistently identifying top-performing policies (or policies within one standard deviation of the best-performing ones).
>
> Another major advantage of SRM over CVaR was observed in the Windy Lunar Lander experiment, where our algorithm demonstrated significantly more stable training. Finding CVaR-optimal policies is known to be challenging in risk-sensitive RL literature ([4]), and we observed this difficulty in our experiments. However, our algorithm, especially with a Mean-CVaR objective, exhibited much more stable training and resulted in the top-performing policy with respect to risk-sensitive metrics.
>
> Finally, as noted in our response to reviewer ymbN, this general risk measure was achieved without sacrificing the interpretability of the optimal policy, which was available for static CVaR-optimal policies. Our approach leverages the decomposition of static coherent risk and the distributional return distribution (Section 5), allowing the policy’s risk sensitivity to be continuously monitored when deployed in real-world settings.
>
> The combination of SRM’s flexibility, the convergence guarantees established in our manuscript, and the interpretability tools presented in Section 5 makes SRM an excellent choice for risk-sensitive policy optimization.
>
> **Other:**
>
> We appreciate the reviewer's feedback on the clarity of our notation. To address these concerns, we will revise the notation in the revised manuscript as follows:
>  - We will use $\boldsymbol{\pi}$ instead of $\boldsymbol{\pi}_{\mathrm{H}}$  to represent the general class of history-dependent policies.
>  - We will use $\mathcal{Y}$ to denote the interval $[G_{\min}, G_{\max}]$, consistent with the notation used in [1].
>
> **References:**
>
> [1] Bäuerle and Rieder. More Risk-Sensitive Markov Decision Processes
>
> [2] Bastani et al. Regret Bounds for Risk-Sensitive Reinforcement Learning
>
> [3] Bellemare et al. Distributional Reinforcement Learning
>
> [4] Greenberg et al. Efficient Risk-Averse Reinforcement Learning

---

> > ### Comment · Reviewer_pk66 · 2025-04-02
> >
> > Thank you for your reply. You have addressed all the questions I raised.
> >
> > I'm sorry that I have an additional question that I forgot to ask before. Since QR-SRM can be regarded as an alternating descent algorithm, and as stated in Theorem 4.1 and Conclusion (iii), QR-SRM does not necessarily converge to the global optimal solution.
> > Then, is it possible to characterize the optimality gap?
> >
> > For example, in the simplest case of CVaR, the baseline algorithm QR-CVaR proposed in Bellemare's book can theoretically guarantee convergence to the optimal solution.
> > In the CVaR case, can we derive theoretical guarantee for QR-SRM?
> >
> > Since this new question is raised during the discussion stage and it may be difficult to answer, I do not insist that the authors provide a perfect reply within the short time. However, I think it would be beneficial to understand this optimality gap theoretically. For example, you could include some references in the paper that discuss the optimality gap of similar problems (e.g. EM algorithm), and discuss the possible methods and conditions for establishing the optimality gap of QR-SRM.
> >
> > Overall, I think this is a good paper, I will raise the score.

---

> > > ### Author Response · Authors · 2025-04-03
> > >
> > > Thank you for this nuanced observation and insightful question.
> > >
> > > A key property of CVaR that enables convergence to the optimal policy in QR-CVaR is that the information required to find the optimal solution can be summarized in a single variable. In the CVaR case, let $q_\alpha$ denote the $\alpha$-quantile in function $h$. Under this formulation, the greedy action selection (Equation 10) simplifies to the following equation, which aligns with the CVaR greedy policy discussed in [1].
> > >
> > > \begin{equation}
> > > a_{G,h}(x, s, c)=\underset{a \in \mathcal{A}}{\arg \max } \mathbb{E}\left[(G(x,s,c,a) - \frac{q_\alpha-s}{c})^{-}\right].
> > > \end{equation}
> > >
> > > Note that in this case, we only need to track a single variable, $\frac{q_\alpha-s}{c}$, rather than all three variables $q_\alpha$, $s$, and $c$, which simplifies action selection. In [1], this variable is denoted by $b$.
> > >
> > > The significance of using a single variable becomes evident at the end of Lemma 7.26 of [1], where finding the initial $b$ requires searching over all possible values of $b$ in $[G_{\mathrm{MIN}}, G_{\mathrm{MAX}}]$. This step is crucial for proving convergence to the optimal solution for CVaR.
> > >
> > > From a theoretical perspective, for SRM, if we extend the state space to include every quantile required to define h, we could perform a similar search. However, this approach is very computationally expensive and impractical. Therefore, we did not adopt this method in our work and instead focused on a more scalable approach that balances theoretical soundness with practical feasibility.
> > >
> > > We agree that a more thorough comparison with the theoretical results in [1] would strengthen the foundation for future research on this topic. We will include a dedicated section in the Appendix to address this. In this section, we will also discuss the optimality gap explored in the literature, such as the one for Entropic Value-at-Risk (EVaR) discussed in [2], to provide a broader perspective on the theoretical properties of risk-sensitive objectives.
> > >
> > > Thank you again for this excellent question. We also sincerely appreciate you raising your score.
> > >
> > > **References:**
> > >
> > > [1] Bellemare et al. Distributional Reinforcement Learning
> > >
> > > [2] Hau et al. Entropic Risk Optimization in Discounted MDPs

---

### Official Review · Reviewer_ymbN · 2025-03-13

**Overall Recommendation:** 3

**Summary:**

This work focuses on the risk-sensitive RL, i.e., maximizing the return and managing worst-case scenarios. As distribution RL considers the return distribution, there are several works applying it to the risk-sensitive RL. Extending the widely used risk metric CVaR to Spectral Risk Measures (SRM), this work proposes a novel safe reinforcement learning method with convergence guarantee. Experiments show that the proposed method outperforms existing risk-neutral and risk-sensitive DRL models in various settings.

**Claims And Evidence:**

yes

**Essential References Not Discussed:**

As this work mainly focuses on risk-sensitive RL, several works closely related to this work are missed, like CVaR [1-4], EVaR [5], and so on.

Reference:

[1] Risk-sensitive and robust decision-making: a cvar optimization approach

[2] Towards safe reinforcement learning via constraining conditional value-at-risk

[3] CVaR-constrained policy optimization for safe reinforcement learning

[4] Risk-sensitive reward-free reinforcement learning with cvar

[5] Risk-sensitive reinforcement learning via Entropic-VaR optimization

**Experimental Designs Or Analyses:**

yes

**Methods And Evaluation Criteria:**

yes

**Other Comments Or Suggestions:**

N/A

**Other Strengths And Weaknesses:**

Strength:

- Extending CVaR to more general metric Spectral Risk Measures (SRM) is novel in risk-sensitive RL

- The proposed method is novel with solid proof.

Weakness:

- It seems that different methods in Table 2 perform better on different indicators. Is there any insights for this phenomenon?

- The experimental scenario is relatively simple. Can the proposed method be applied to more complex tasks, such as continuous robot control?

- Minors: The spacing between rows in some places looks very small and needs to be adjusted

**Questions For Authors:**

See weaknesses above

**Relation To Broader Scientific Literature:**

This work is established on previous risk-sensitive RL methods based on CVaR and introduces Spectral Risk Measures (SRM) as a more general metric.

**Theoretical Claims:**

roughly

---

> ### Author Rebuttal · Authors · 2025-03-31
>
> Thank you for your positive feedback and for recognizing the strengths of our work. We are glad that you found our proposed method novel. We address your questions below.
>
> **Missing References:**
>
> Thank you for highlighting the missing references. The first reference is already discussed in the paper, but we will ensure the remaining references are added in the updated version.
>
> **Strengths and Weaknesses:**
>
> Thank you for recognizing the strengths of our work. We would like to emphasize that extending static CVaR to the more general static SRM is just one of our contributions. Another significant contribution is the interpretation of the learned policy through the decomposition of static coherent risk measures and the distribution of returns in our algorithm, as discussed in Section 5. Unlike existing works in the risk-sensitive RL literature, our approach introduces a decomposition of risk preference that enables tracking the agent's intermediate, state-dependent risk preferences. This added interpretability is a unique feature of our method, allowing for continuous monitoring of the policy's behavior. If the policy ever diverges from the user's preferences, a new policy can be trained to realign with those preferences.
>
> **Q1:**
>
> While different methods in Table 2 excel on different indicators, our proposed algorithm (QR-SRM) consistently ranks as the top-performing or within one standard deviation of the top-performing algorithms across all objectives. These minor variations can be attributed to factors such as the inherent stochasticity of the environment and the use of function approximation for value functions. We will highlight the performance of our model more clearly in the revised manuscript to further demonstrate its strong performance.
>
> **Q2:**
>
> As mentioned in the conclusion section, our value-based method is designed for environments with discrete action spaces. Extending our algorithm to actor-critic methods would enable its application in environments with continuous action spaces. In fact, addressing the continuous control problem is the focus of our upcoming work.
>
> **Q3:**
>
> Thank you for bringing this to our attention. We will address this issue in the revised draft.

---

> > ### Comment · Reviewer_ymbN · 2025-04-02
> >
> > I have read the rebuttal and other reviews, and I keep my score that this paper is solid as well as novel.

---

> > > ### Author Response · Authors · 2025-04-03
> > >
> > > Thank you for taking the time to read our rebuttal and the discussion. We truly appreciate your thoughtful engagement and for recognizing our work as solid and novel.

---

### Decision · Program_Chairs · 2025-05-01

**Decision:**

Accept (poster)

**Comment:**

The reviewers agree that the paper presents a valuable contribution to risk-averse RL. The paper presents new insights into a complex problem and new general algorithms. The results in the paper are not surprising, and the paper builds on several recent papers that introduce and study similar ideas. In fact, the reviewers point out recent work that introduced similar results for spectral risk measures. The authors, however, convincingly argue that the contribution compared with this prior work is significant. In addition, the reviewers uncovered a few relatively minor technical issues. The author's response indicates that these problems are quite easy to address in the final version of the paper.